# Wafer-scale functional circuits based on two dimensional semiconductors with fabrication optimized by machine learning

Xinyu Chen[1,7], Yufeng Xie[1,7], Yaochen Sheng[1,7], Hongwei Tang[1,7], Zeming Wang[1], Yu Wang[1], Yin Wang[1], Fuyou Liao[1], Jingyi Ma[1], Xiaojiao Guo[1], Ling Tong [1], Hanqi Liu[1], Hao Liu[1], Tianxiang Wu[1], Jiaxin Cao[1], Sitong Bu[1], Hui Shen[1], Fuyu Bai[1], Daming Huang[1], Jianan Deng [2], Antoine Riaud [1], Zihan Xu[3], Chenjian Wu[4], Shiwei Xing[4], Ye Lu[2], Shunli Ma[1], Zhengzong Sun [1], Zhongyin Xue[5], Zengfeng Di[5], Xiao Gong[6], David Wei Zhang[1], Peng Zhou [1✉], Jing Wan[2✉] & Wenzhong Bao [1✉]

Triggered by the pioneering research on graphene, the family of two-dimensional layered materials (2DLMs) has been investigated for more than a decade, and appealing functionalities have been demonstrated. However, there are still challenges inhibiting high-quality growth and circuit-level integration, and results from previous studies are still far from complying with industrial standards. Here, we overcome these challenges by utilizing machine-learning (ML) algorithms to evaluate key process parameters that impact the electrical characteristics of MoS$_2$ top-gated field-effect transistors (FETs). The wafer-scale fabrication processes are then guided by ML combined with grid searching to co-optimize device performance, including mobility, threshold voltage and subthreshold swing. A 62-level SPICE modeling was implemented for MoS$_2$ FETs and further used to construct functional digital, analog, and photodetection circuits. Finally, we present wafer-scale test FET arrays and a 4-bit full adder employing industry-standard design flows and processes. Taken together, these results experimentally validate the application potential of ML-assisted fabrication optimization for beyond-silicon electronic materials.

[1] State Key Laboratory of ASIC and System, School of Microelectronics, Fudan University, Shanghai 200433, P. R. China. [2] State Key Laboratory of ASIC and System, School of Information Science and Technology, Fudan University, Shanghai 200433, P. R. China. [3] Shenzhen Six Carbon Technology, Shenzhen 518055, P. R. China. [4] School of Electronic and Information Engineering, Soochow University, Suzhou 215006, P. R. China. [5] State Key Laboratory of Functional Materials for Informatics, Shanghai Institute of Microsystem and Information Technology, Chinese Academy of Sciences, 865 Changning Road, Shanghai 200050, P. R. China. [6] Department of Electrical and Computer Engineering, National University of Singapore, Singapore 117583, Singapore. [7] These authors contributed equally: Xinyu Chen, Yufeng Xie, Yaochen Sheng, Hongwei Tang. ✉email: pengzhou@fudan.edu.cn; jingwan@fudan.edu.cn; baowz@fudan.edu.cn

Two-dimensional (2D) semiconductors have potential applications from mainstream logic and analog circuits to flexible electronics[1–8]. Semiconductive transition-metal dichalcogenides (TMDs) are a family of 2D semiconductors with versatile band structures, among which MoS$_2$ is the most widely studied representative of TMDs[9–18]. The atomically thin channel with dangling-bond-free interfaces and low in-plane dielectric constants ensures high carrier mobility in extremely scaled devices with robust control over short-channel effects (SCEs)[19–21]. While intrinsic advantages of 2DLMs are promising for more-than-Moore electronic applications[22–25], it is still challenging to meet the stringent requirements for large-scale circuit- and system-level applications, where the primary challenges are wafer-scale material synthesis and device processing[26–33]. Recently, worldwide research efforts on chemical vapor deposition (CVD) and metal–organic CVD synthesis have enabled semiconductive TMD films with large areas[34–36]. Although satisfied crystalline quality and large-scale uniformity still require further improvement of synthesis techniques, currently available wafer-scale TMD films are practically sufficient for fabricating large-scale circuits.

In order to realize complex cascaded circuits based on 2D semiconductors, voltage-level matching and high noise margins are also important[37], placing the need for the accurate control of threshold voltage ($V_T$) of field-effect transistors (FETs). So far, a functional circuit consisting of 115 MoS$_2$ FETs fabricated by a gate-first technology has been reported[37,38]. However, such gate-first technology requires a more complex film-transfer processing and an extra step to form contact via, which not only introduces defects to MoS$_2$ films but also drastically reduces the yield and reproducibility of wafer-scale fabrication. Moreover, from a practical point of view, a top-gate (TG) structured FET with a high-k dielectric layer (i.e., conventional gate-last technology), is necessary for independent gate control and circuit-level integration[39]. Hence, large-scale circuits require more emphasis on TG-FET fabrication optimization toward wafer-scale uniformity and reproducibility. However, the ultrathin nature of 2D semiconductors makes them extremely sensitive to exterior environments and fabrication processing, especially the top interface of 2D semiconductors. In their TG-FET fabrication procedure, all individual processing steps are highly coupled to each other because any subsequent processing steps will influence the previous ones, making the processing optimization of 2D semiconductors more complicated than those in bulk semiconductors such as Si and Ge.

In this work, to realize batch fabrication using 2-in. MoS$_2$ wafer, machine-learning (ML) algorithms were used to analyze experimental data and evaluate various key process parameters that significantly impact the electrical characteristics of 2D-FETs, enabling optimized electrical performance for enhancement-mode FETs fabricated using ML-guided gate-last processing. Calibrated by measured electrical data, the device modeling is conducted to guide the design of basic digital, analog, and optoelectrical circuits. With wafer-scale processing using industry-standard design flows and processes, our work illustrates the feasibility of using ML in device-processing optimization for emerging novel materials and shortens the learning cycle from fundamental research to practical application.

## Results

**Machine learning-assisted co-optimization**. High-quality, uniform MoS$_2$ was grown using customized CVD equipment (see Methods). Raman mapping results indicate that the synthesized wafer is uniform at the wafer scale, as shown in Fig. 1a (see Supplementary Note 1 for more details). The subsequent fabrication of high-performance MoS$_2$ FETs requires optimizing individual processing modules, such as channel doping, source–drain contacts, and TG gate stack. Due to the extremely high sensitivity of carriers in the MoS$_2$ channel to the ambient environment, these processing steps are strongly coupled together through the MoS$_2$ channel interface and an ultrathin TG dielectric layer (around 10-nm thick), making comprehensive process optimization much more complex and challenging, as illustrated in Fig. 1b. The processing steps are all correlated to the final device-performance metrics, including carrier mobility ($\mu$), threshold voltage ($V_T$), subthreshold swing (SS), and current on–off ratio ($I_{on}/I_{off}$), as shown in Fig. 1c. For practical applications, it is necessary to optimize the combination of these quantities, and different device applications also require different optimization strategies, e.g., a high $\mu$ is critical for faster operation speed, and a small SS is essential for low-power consumption. After optimizing wafer-scale material and device-fabrication processes, we can continue the device characterization, SPICE modeling, and circuit design. The obtained device and circuit-characterization results can also be further used to guide improvements to the fabrication process, as illustrated by Fig. 1d.

The fast-developed ML technology is commonly used for the efficient understanding of complex mathematical or logical models. ML has been used in many disciplines, such as exploring novel materials[40], but there has never been any report on using ML to optimize process modules for 2D devices. Here, we show that ML can improve the fabrication process of devices built on emerging semiconductors more effectively than the conventional process-optimization method. Specifically, ML is used to understand the impact of each processing step on the final device performance. This is essential for materials, such as MoS$_2$ grown via CVD on an insulating substrate, making device measurements after each processing step difficult.

A complete process for fabricating MoS$_2$ TG-FETs is schematically shown in Fig. 2a (also see Supplementary Note 2 for detailed processing steps). The FET performance is measured at the end of the process flow. Ensemble learning (EL), a supervised ML method where multiple learning algorithms are aggregated for more accurate prediction[41], is used here as it is effective for classifying imbalanced data (details see Supplementary Note 3). The decision-tree method is used as a weak classifier because it can efficiently handle discrete data (Fig. 2b). More than 560 MoS$_2$-FETs on over 40 different wafers were fabricated using specially designed process flows to provide a comprehensive database. We first focus on two device-performance parameters, $\mu$ and $V_T$, as $\mu$ is directly correlated to operation speed and $V_T$ is essential for fabricating an enhancement-mode FET. The importance of each processing step can be determined using one favored parameter ($\mu$ or $V_T$) as the sorting standard for EL analysis (Fig. 2c). The generated results are reasonable upon physical analysis, since $V_T$ is primarily influenced by the TG structure (metal work function and charge impurities/dipoles in the deposited-gate dielectric). At the same time, the mobility $\mu$ is extracted by the Y-function method, which depends on multiple factors such as interfacial scattering and contact resistance[42]. The TG-electrode metallization also becomes an essential step as indicated by ML analysis, which is unexpected (for details see Supplementary Note 7). $\mu$, $V_T$, and other performance parameters can be comprehensively considered by multiplying a weighting factor for each parameter, depending on the requirements of various functionalities.

We then demonstrate that ML can also be used to co-optimize all process steps, as shown in Fig. 2d. After the EL training, a score predictor can predict the results from a specific processing combination (i.e., one process recipe). All possible process recipes

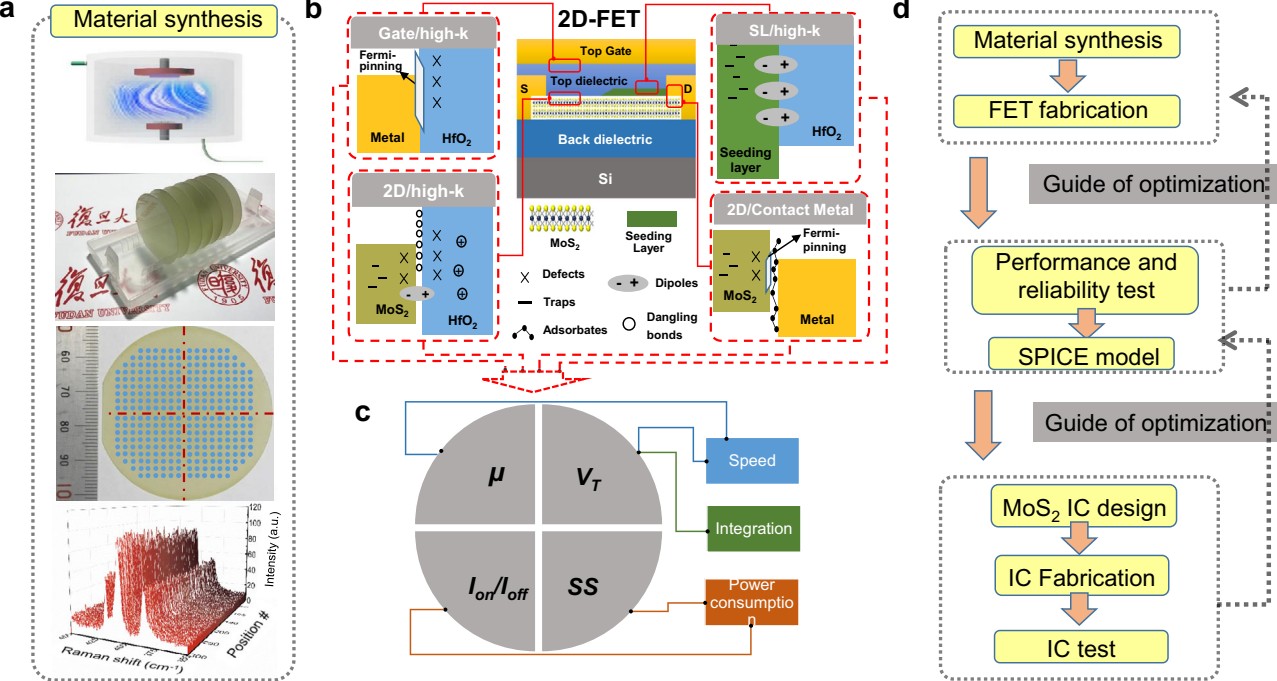

**Fig. 1 A comprehensive picture of building MoS$_2$ 2D-FETs. a** Demonstration of uniform wafer-scale MoS$_2$ growth by CVD, including a schematic diagram of the material-growth equipment, a batch of 2-in. wafer-scale sapphire substrates uniformly covered with MoS$_2$, a 2-in. sapphire wafer uniformly covered with MoS$_2$ marked with Raman test points, and Raman mapping spectra from different locations marked in the previous picture. **b** Schematic cross section of an MoS$_2$ FET with TG (top gate) and global BG (bottom gate). Various interface factors that influence the device performance are categorized, including the insertion of seeding layer (SL) between MoS$_2$ and high-k dielectric (in this work, SL is deposited on the entire channel region), the interface between the TG and high-k dielectric, and the interface between MoS$_2$ and contact metals. **c** Schematic diagram of the relationship between performance parameters of the transistor and performance limitations of the integrated circuit, where $\mu$, $V_T$, $I_{on}/I_{off}$, and SS represent the mobility, threshold voltage, current on/off ratio, and subthreshold swing. **d** Process flow and feedback-optimization diagram from material synthesis to industrial-grade circuit design, fabrication, and test.

are then sorted using a grid-search method, as shown in Fig. 2e. To demonstrate this, we fabricated more than 500 MoS$_2$ FETs, which are summarized in the $\mu$–$V_T$ plot in Fig. 2f. Each color corresponds to FETs fabricated by one process recipe. Most recipes were designed by human experiences based on step-by-step optimization (details see Supplementary Notes 4-9). For example, one recipe provides a high $\mu$ value (orange circles), and another provides a positive $V_T$ (blue circles). However, mixing two recipes (green circles) cannot guarantee both high $\mu$ and positive $V_T$, mainly due to crosstalk between different processing steps (for detailed discussion see Supplementary Note 9). Therefore, the combination of multiple steps with each optimized does not necessarily generate the best device. We then fabricated a batch of devices (red stars in Fig. 2f) following the suggestion of the sorting result (red arrow in Fig. 2e). This recipe (processing details, see Supplementary Note 9) also gives rise to an average $\mu$ of about 75 cm$^2$/V·s and $V_T$ of 2.1 V, as well as a high wafer-scale uniformity that is important for large-scale circuits, as shown in Fig. 2g (see more electrical characterizations in Supplementary Note 10). In the future, device physics is still necessary to understand each aspect deeply for further optimization. However, the detailed physical explanations are not the focus of this work.

Therefore, compared with the traditional design of experiment (DoE), our ML-assisted approach can effectively reduce the research workload of complex co-optimization. Here, the application of the ML algorithm for MoS$_2$ TG-FET optimization is only a case study, and its capability to reduce the learning cycle of device optimization can be conveniently extended to other emerging electronic materials and novel devices.

**From transistors to circuits**. Since the FETs built on the wafer have high uniformity, we use an RPI model (level = 62) to simulate MoS$_2$ FETs in an HSPICE simulator. As is shown in Fig. 3a, b, to fit the transfer and output characteristics of MoS$_2$ FETs, the parameters of the model are configured by adjusting the empirical parameters and characteristic parameters (such as mobility and $V_T$ extracted from transfer curves, thickness and permittivity of the dielectric, and $W$ and $L$ of MoS$_2$ channel). The voltage-transfer characteristics (VTC) for a pseudo-NMOS MoS$_2$ inverter (M1 as a load transistor and M2 as a pull-down network) were also simulated in HSPICE using the simulation parameters from the same model. By sizing the aspect ratio $W/L$ of two MoS$_2$ FETs (Fig. 3c) and shifting the $V_T$ value (Fig. 3d) of the M1 independently, the voltage-switching point can be tuned to the proper position (around half of $V_{DD}$) to achieve rail-to-rail output swing and large noise margin (Supplementary Note 11).

A flip-flop is a fundamental storage element for sequential ICs[43–46]. Figure 3e shows a circuit schematic and a die photo of a negative edge-triggered D flip-flop (DFF) based on 8 NANDs with 2 inputs and 3 inverters. The measured waveforms from the DFF are plotted in Fig. 3f, where the device outputs correct logic values for given input data on the falling edge of the clock (CLK) and holds the data until the next falling edge. A full adder is another key combinational circuit usually used as a fundamental building block in an arithmetic logic unit (ALU)[37,47]. Figure 3g shows a circuit schematic of a 1-bit full adder and a photograph of the die. The 1-bit full adder consists of 10 NANDs, three inverters, and 1 NOR with 39 n-FETs in total. The measured-output waveforms from the 1-bit full adder are shown in the

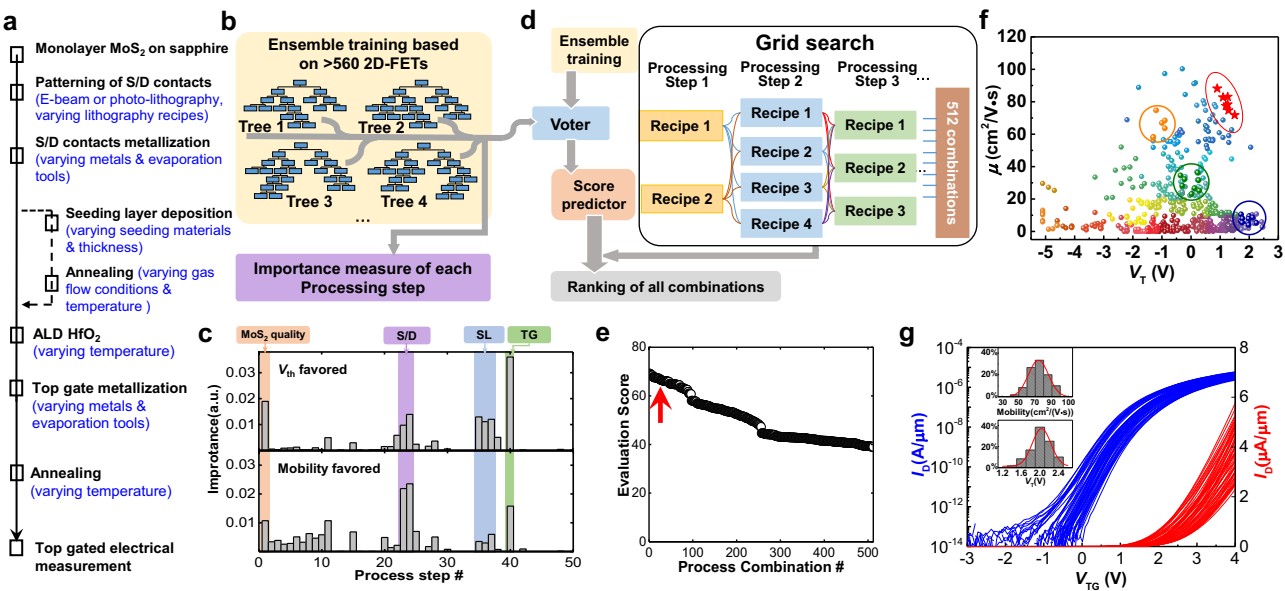

**Fig. 2 Machine learning assisted optimization of MoS₂-device process. a** Process flow for fabricating TG MoS₂ FETs. The variations in each step are marked in blue. **b** Graphical representation of ensemble learning (EL) based on decision-tree algorithm. The importance of each processing step is extracted during the creation of decision trees. **c** Importance of processing steps for $\mu$ and $V_T$ based on random-forest regression, where S/D, SL, and TG represent the source/drain contacts, the seeding layer, and the top gate. **d** Diagram of co-optimization procedure based on ML. After training with EL, a score predictor can predict the overall device performance for all processing combinations using a grid-search method. **e** Ranking of all possible processing combinations. The high-score combinations can be referenced for device fabrication. **f** More than 500 MoS₂ TG-FETs summarized in a $\mu$–$V_T$ plot. Each color corresponds to one process recipe. The red stars are the results of the process recipe in **e** pointed by the red arrow. The orange, green, and blue points are three batches of devices fabricated for a control experiment discussed in the maintext. **g** Transfer characteristics for 60 MoS₂ TG-FETs on one wafer at $V_{DS} = 0.5$ V in linear and logarithmic coordinates. The insets show histograms and Gaussian fits (red solid lines) of statistical data for Y-function-calculated mobility (upper) and threshold voltage (downside) to Gaussians.

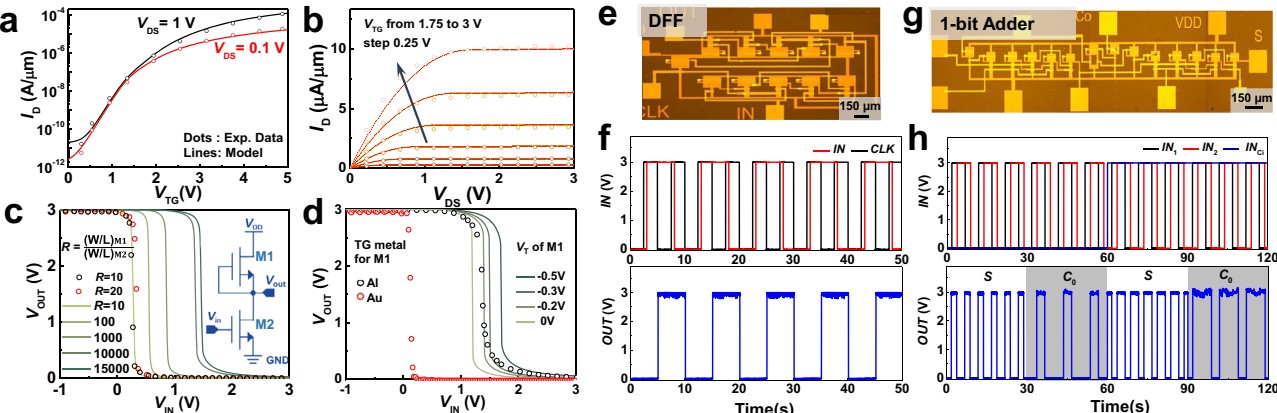

**Fig. 3 Logic circuits based on MoS₂ TG-FETs.** Experimental data (circular dots) and simulation (lines) for **a**, **b** transfer and output characteristics of MoS₂ TG-FETs, and **c**, **d** display voltage-transfer curves (VTCs) of an MoS₂ inverter with M1 and M2 FETs. The inset in **c** is the schematic of a MoS₂ pseudo-NMOS inverter. The geometry parameter $R = (W/L)_{M1}/(W/L)_{M2}$ is used to adjust the switching point of the VTC curve in **c**, while a different method is used in **d** by independently tuning $V_T$ of M1. **e** is an optical microscope image of a MoS₂-negative edge-triggered D flip-flop (DFF) circuit, and **f** shows the corresponding experimental results. The upper two waveforms are inputs with a 0–3 V voltage swing, and the lower graph shows the measured output. **g** is an optical microscope image of a 1-bit MoS₂ full adder and **h** is the corresponding experimental results. The output signal of sum (S) and carry output ($C_o$) is distinguished by the shaded gray areas.

bottom plots of Fig. 3h, where the outputs ("S" and "Co") produce the correct rail-to-rail voltage for all possible input combinations with 3.0 V supply voltage. More logic modules are also demonstrated in Supplementary Notes 12–13.

A ring oscillator (RO) is an industrial standard benchmarking circuit for performance evaluation[47,48]. We then fabricated and measured a 5-stage pseudo-CMOS RO with an output buffer

(Fig. 4a) to assess the high-frequency switching capability of MoS₂. Such RO circuit is composed of five inverters cascaded in a loop chain. High uniformity of all inverter stages, such as their large noise margin, is essential for robust oscillator performance. As shown in Fig. 4b, an oscillation frequency of 19.5 kHz with a propagation delay of $\tau_{pd} = 1/(2nf) = 5.13\,\mu s$ per stage was measured at $V_{DD} = 3$ V, where $n$ is the number of stages. The

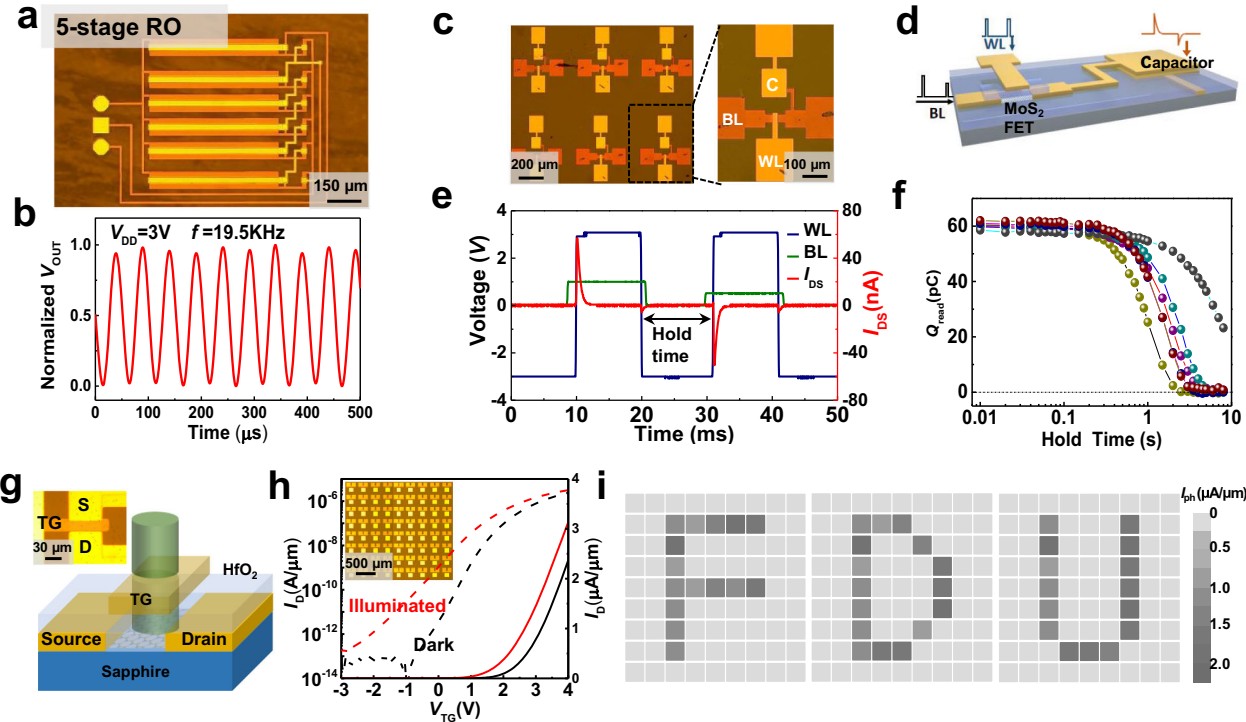

**Fig. 4 Analog, memory, and optoelectronic circuits based on MoS₂ TG-FETs. a** is an optical microscope image of a 5-stage ring oscillator, and **b** is the corresponding output characteristics at 19.5 kHz with $V_{DD} = 3$ V. **c** Optical microscope image of MoS₂ memory-unit arrays. The right zoom-in shows the detailed structure of a 1T-1C dynamic memory circuit, whose schematic diagram is shown in **d**. **e**, Write-and-read operations in the 1T-1C unit. WL, BL, and $I_{DS}$ represent the write line, bit line, and working current. **f** Calculated charge stored in the capacitor as a function of holding time for five different devices. **g** Schematic diagram of an MoS₂ phototransistor with a 10-nm-thick Au top gate, and **h** displays transfer characteristics with and without illumination at $V_{DS} = 0.5$ V. The insets in **g**, **h** are the optical microscopic images of a MoS₂ phototransistor and its large-scale arrays, respectively. **i** Photocurrent mapping for a 9 × 9 MoS₂ FET array. The photocurrent is produced by scanning the array using a microscope-focused white beam.

self-oscillation frequency of our RO is relatively low compared with previous reported results[11] (for discussion see Supplementary Note 14), but there is a large room for future improvement via downscale of device size.

For memory applications, we present dynamic memory arrays built from MoS₂ FETs (Fig. 4c). A schematic diagram of a 1T-1C circuit is shown in Fig. 4d. An oscilloscope was used to test its function as memory (Supplementary Note 15)[49–53]. The experimental results are shown in Fig. 4e. During a write operation, the MoS₂ FET is turned on to provide a low-impedance path, and a positive current pulse (red curve) is collected by the oscilloscope, which indicates the capacitance has been recharged. During the holding state, the MoS₂ FET is turned off and presents a high-impedance path. If the current pulse detected by the oscilloscope is negative during a read operation, it indicates that a charge remains in the capacitor after the hold time. Due to the ultralow leakage current from our MoS₂ FET, the charge saved in the capacitor is expected to be ideally stored, thereby achieving long-term retention. By integrating the current pulse during a read operation, we can estimate the charge retained in the capacitor as a function of hold time, as shown in Fig. 4f. The retention time is defined as the hold time at which the retained charge ($Q_{read}$) is zero compared with a read voltage of 0.5 V; the average retention time is on the order of seconds (Supplementary Figure 18).

Furthermore, our wafer-scale MoS₂ devices can be extended for optoelectrical application[54,55]. A thin layer Au (~10 nm) deposited as TG electrode will have higher optical transmittance, as shown in Fig. 4g. The transfer characteristics (Fig. 4h) from a typical device indicate a photocurrent of ~1 µA/µm under white light (1.5 mW/cm²) when $V_{TG} = 4$ V and an on–off ratio of

approximately 100 when $V_{TG} = 0$ V (more details see Supplementary Note 16). In Fig. 4i, we use a 9 × 9 MoS₂ FET array to demonstrate a simple function of image sensing. The photocurrents are recorded from each pixel by scanning a focused white beam across the array. We set the illumination position to form the English letters F, D, and U deliberately. The color pattern representing the photocurrent value exhibits high on/off contrast and high spatial uniformity.

Here we have demonstrated logic, analog, memory, and optoelectronic functions, which can be conveniently integrated into a single device. In the future, if we further take advantage of the atomically thin and flexible nature of 2D materials, it is possible to prepare three-dimensional monolithic integrated circuits (3D integration) by stacking 2DLMs with different functions[56]. Thus, it provides a new route to implement a complex system to realize various applications.

**Wafer-scale fabrication.** To demonstrate the potential for high-volume production, we fabricated MoS₂ TG-FET arrays and 1-bit full-adder arrays on a 2-inch wafer, as shown in Fig. 5a. Similar to what is normally completed in a semiconductor-fabrication facility, the full-adder arrays were placed in the center region of the wafer as a functional block, and MoS₂ TG-FETs were placed surrounding the functional blocks and used to monitor wafer-scale uniformity. Each block contained 16 FETs, and 81 blocks in total were distributed across the wafer. The average mobility and $V_T$ values extracted from the transfer curves in each FET array are plotted in Fig. 5b, showing a wafer-scale uniformity acceptable for batch fabrication. The average mobility and $V_T$ values for all 1296

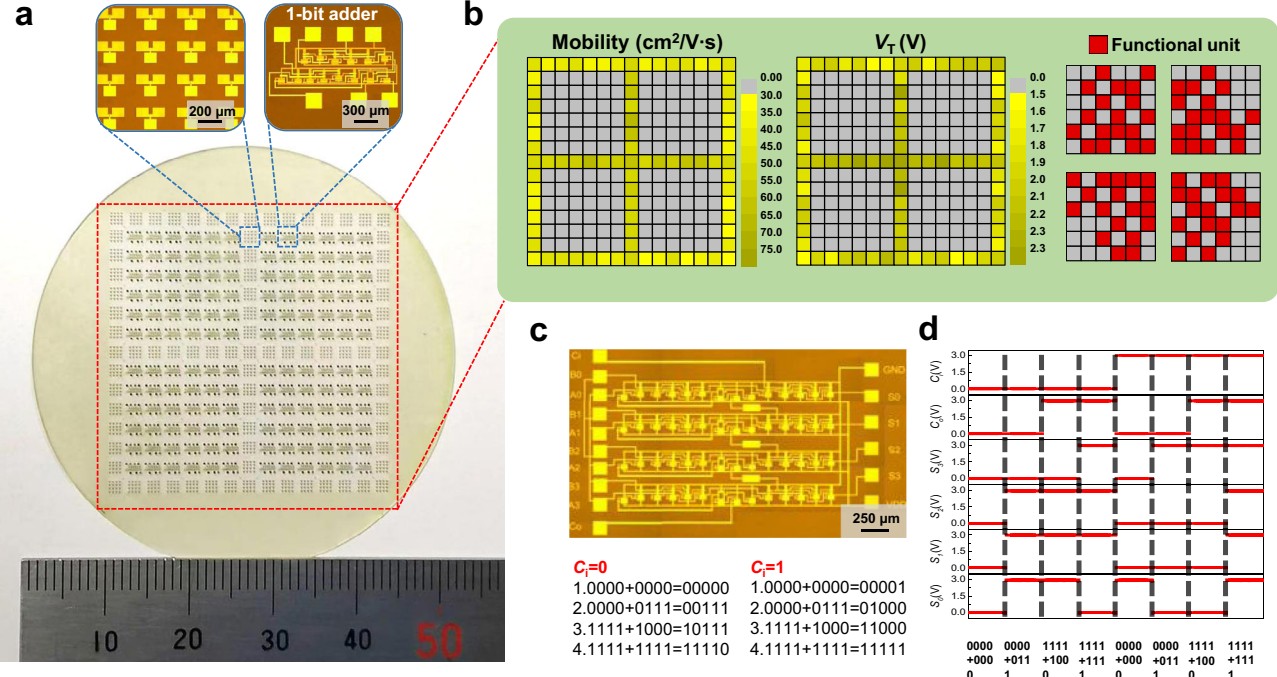

**Fig. 5 Wafer-scale integrated circuits built from MoS₂ FETs. a** Photograph of a 2-inch MoS₂ wafer with 1-bit full-adder arrays as functional circuits in the center, and MoS₂ TG-FET arrays used as monitoring devices locating in the surrounding regions. All MoS₂ TG-FETs and 1-bit full-adders are fabricated within the red dashed square. The zoom-ins of blue dashed boxes are optical microscope images of the corresponding MoS₂ TG-FET and 1-bit full-adder arrays. **b** Wafer maps of mobility (left) and $V_T$ statistics (center) extracted from devices in the surrounding regions. The yellow scale bars show mobility and $V_T$ values. Each block's color scale represents a value averaged from 16 FETs, and the entire wafer has 81 blocks. The right graph illustrates the yield of 1-bit full-adder circuit arrays. The red and gray squares represent the proportion of working and nonworking circuits, respectively. **c** Optical microscope image of a 4-bit full adder under which is the truth table for logical combinations. **d** Functional measurements of the 4-bit full adder with $V_{DD} = 3$ V. The 4-bit full-adder was tested using a series of input combinations (A, B) in the following order: (0000 + 0000, 0000 + 0111, 1111 + 1000, 1111 + 1111) with $C_i = 0$ and $C_i = 1$. The y axes are the voltage of carry input ($C_i$) and carry output ($C_o$), and the output voltage of sum₀ ($S_0$), sum₁ ($S_1$), sum₂ ($S_2$), sum₃ ($S_3$).

MoS₂ FETs are 46.7 cm² V⁻¹ s⁻¹ and 1.9 V, respectively, with a standard deviation <30%. In the rest of the wafer area, we tested 144 1-bit full-adder circuits, revealing a yield of about 50% (right graph of Fig. 5b, and more discussion see Supplementary Note 17). These results indicate that our wafer-scale MoS₂ film, together with optimized device-processing technologies, can potentially achieve industrial high-volume production. To the best of our knowledge, these are among the highest-mobility and $V_T$ values observed in wafer-scale-fabricated MoS₂ TG devices with high uniformity (for a detailed comparison see Supplementary Note 18). Finally, we fabricated a complete 4-bit full adder composed of four parallel 1-bit full adders consisting of 156 FETs; the microscope image and truth table are shown in Fig. 5c. The 4-bit full adder was tested using eight input-signal combinations (A3 A2 A1 A0, B3 B2 B1 B0, Ci), including (0000, 0000, 0), (0000, 0111, 0), (1111, 1000, 0), (1111, 1111, 0), (0000, 0000, 1), (0000, 0111, 1), (1111, 1000, 1), and (1111, 1111, 1). The output results in Fig. 5d show that the 4-bit full adder exhibits correct logical function and rail-to-rail conversion. Thus, we have demonstrated that our ML-guided MoS₂ fabrication technology provides a potential route for constructing future large-scale 2D ICs compatible with current silicon-based technologies.

## Discussion

The synthesis of wafer-scale MoS₂ and other 2D semiconductors is currently under fast development, providing more material candidates for fabricating FETs and ICs. Even for the MoS₂ film itself investigated in this work, the synthesis method can be further optimized to modify the grain size, crystallinity, defect density, etc.[57], which all influence the overall performances of the

MoS₂ FETs. This is one of the main reasons why academic researchers have opted not to undertake strenuous efforts on the fabrication optimization of specific 2D semiconductors. Therefore, our results can be extended to other 2D semiconductors and emerging novel materials to reduce their device-optimization burdens and shorten the learning cycle. Of course, such a speed-up approach is more suitable at the initial phases in device optimization. Once a certain level is reached, the understanding of device physics is still needed for further improvement.

## Methods

**Synthesis of wafer-scale MoS₂.** A crucible with MoO₃ power (Alfa Aesar 99.95%) is placed in Zone 2, and an appropriate amount of sulfur powder (Alfa Aesar 99.999%) is placed in Zone 1, which is upstream of the flow in the tube. The distance between the two zones is 30 cm. A carefully rinsed sapphire substrate is placed face-down on the MoO₃ power. During the synthesis process, 300-sccm argon gas serves as a carrier gas. The synthesis temperature for Zone 1 and Zone 2 is controlled at 180 °C and 650 °C, respectively. A continuous-monolayer MoS₂ film is synthesized at atmospheric pressure with 10 min of sulfuration time.

**The machine-learning method.** The details of ensemble learning, random-forest algorithm, and feature-importance assessment are described in Supplementary Note 3.

**Overall fabrication procedure of MoS₂ FETs and circuits.** The MoS₂ FETs and circuits are fabricated on the wafer-scale MoS₂ film on the sapphire substrate. The contact electrodes, source and drain contacts are patterned by laser direct writing technology (Micro-Writer ML3) and subsequently deposited using electronic beam (E-beam) evaporation. CF₄ plasma etching is performed to define a MoS₂-channel region. A seeding layer is deposited by E-beam evaporation and subsequently annealed in an oxygen atmosphere at 100 °C. Then HfO₂ layer was grown by atomic-layer deposition (ALD) as a high-k TG dielectric layer. Another lithography/lift-off/deposition process is utilized to form the TG metal layer. For electrical probing or further fabrication of more complex circuits, SF₆ plasma

etching removes the $HfO_2$ layer on top of the source/drain electrodes to form via holes defined by the lithography. More fabrication details can be found in Supplementary Note 2.

**Electrical measurement**. The electrical properties of $MoS_2$ FETs and circuits are carried out in a probe station connecting to an Agilent B1500A semiconductor analyzer with eight source-measure units (SMUs). To investigate the circuit's dynamic response, an Agilent 33622 A arbitrary-waveform generator is used to input signals, while a RIGOL DS1054Z digital oscilloscope and an Agilent B1500A semiconductor analyzer capture the output signal.

## Data availability
The datasets generated during and/or analyzed during the current study are available from the corresponding authors upon reasonable request.

## Code availability
The codes used for simulation and data plotting are available from the corresponding authors upon reasonable request.

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

## Acknowledgements

We thank Prof. He Tian for the insightful discussion. This research is supported in part by the National Key Research and Development Program (2016YFA0203900), Innovation Program of Shanghai Municipal Education Commission (2021-01-07-00-07-E00077), Shanghai Municipal Science and Technology Commission (18JC1410300, 21DZ1100900), Shanghai Rising Star Program (19QA1401100), and National Natural Science Foundation of China (61925402, 51802041, 51925208, 61904032, 61874154, 61874031).

## Author contributions

W.B., J.W., and P.Z. led the project. Y.X., Z.W., J.C., and Y.W. developed the ML algorithms. X.C., Y.S., H.T., Y.X., T.W., C.W., S.X., and S.M. contributed to circuit design. X.C., Y.S., H.T., Y.W., F.L., J.M., X.G., and L.T. fabricated the devices and circuits. X.C., H.T., Y.W., H.L., J.D., S.B, H.S., F.B., and D.H. contributed to the electrical measurements. Zih.X., Z.S., Z.X., Z.D., and Hanq.L. prepared the 2D semiconductors. Y.L., X.G, and J.W. advised the wafer-scale circuit design and test. A.R. and D.W.Z. discussed the results. All authors commented on the paper.

## Competing interests

The authors declare no competing interests.
