## [Peer Review File · Nature Communications]

REVIEWER COMMENTS

Reviewer #1 (Remarks to the Author):

The manuscript by Xinyu Chen et al. reports on the investigation of the key process parameters that impact the electrical characteristics of MoS₂ TG-TFTs by utilizing machine-learning (ML) algorithms. The authors fabricated enhancement-mode MoS₂ TG-FETs using ML-guided gate-last processing recipes and analysed them by 62-level SPICE modelling. They were able to demonstrate basic logic, analogue, and optoelectrical circuits. The results show that ML can be used for device optimization and shortening the learning cycle for novel 2D materials. We would like to congratulate the authors on the nice results. The manuscript indicates the very well planned and nicely organised experiments. The paper is clearly structured and well written. The conclusions are supported by the obtained results. We would like to draw attention to only a few points.

1. It is very common to perform transistor measurements in forward-and-back sweeping modes, which allows revealing the hysteresis. The latter can be a detrimental factor for the performance of the electrical circuits. It would be beneficial to discuss this matter and demonstrate the results by corresponding measurements.
2. Throughout the report, the authors investigate a few key parameters for the performance of the transistors, such as mobility and threshold voltage. One of the important factors especially for extremely scaled transistors is contact resistance. Did the authors investigate the influence of the process parameters on the contact resistance?
3. The results reveal a rather low frequency of the ring oscillator self-oscillations. Earlier results give values in the MHz region, which is two orders of magnitude larger than obtained in this work (<https://doi.org/10.1021/nl302015v>). The investigation of the low operating frequency may give insights into further improvement of the circuits.
4. Most of the measurements were performed in either DC mode or at a very low frequency. It seems important to discuss the possible operating frequencies and the limiting factors.
5. The optoelectrical applications were demonstrated by utilizing photodiodes. The authors demonstrate a high ON/OFF ratio. However, it is important to discuss the mode of operation (CW vs pulsed). It is well known that both these modes differ in such parameters as responsivity and ON/OFF ratio <https://doi.org/10.1021/nl502339q>
On the other hand, sensing applications require high-frequency operation and thus it is important to explicitly mention that the measurements were acquired in low-frequency mode.
6. To make the comparison in table S7 more complete, it is worth to mention another recent work on IC on MoS₂ <https://www.nature.com/articles/s41928-020-0460-6>
Does the table state the average or maximum mobilities? I believe it makes sense to explicitly mention it.
7. The yield of a 1-bit full-adder with 39 n-FETS is 50%. It is worth to discuss the reasons for the failures in non-working devices and possible routes to overcome these issues.

Best regards,
Dmitry Polyushkin.

Reviewer #2 (Remarks to the Author):

****THIS REPORT WAS WRITTEN IN COLLABORATION WITH REFEREE #4**

The manuscript "Wafer-Scale Functional Circuits Based on Two Dimensional Semiconductors with Fabrication Optimized by Machine Learning" by Xinyu Cheng and co-authors describes a Machine Learning based process optimization method. This this method was used to optimize the performance of MoS₂ channel field effect transistors. After the authors identified the optimized

process, it was used to fabricate different circuits and to realize wafer scale MoS₂ device fabrication. While it is an interesting approach, the paper does not convincingly describe the advantage of the approach compared to conventional process development. In particular, it is not clear where the ML-based pattern recognition provides an advantage over classic design of experiment guidelines. I therefore think this manuscript is not appropriate for publication in Nature Communication.

Below are my detailed comments.

1. What is the advantage of the process optimization based on ML, over the traditional way (in which we optimize each step and then combine whatever we would like to have for a specific application). The traditional way of designing an experiment even appears to be more efficient, since one may not need to go through all fabrication process possibilities. In a clean DOE, one can run specific process combinations and just by looking at the data identify the most promising route. In the current form of the manuscript, this aspect is not discussed at all.

2. It is not clear for me what this statement indicates: "This is essential for materials, such as MoS₂ grown via CVD, which are synthesized on an insulating substrate, making device measurements after each processing step difficult." The authors should explain in more detail what they mean with this statement for the reader, as there are many established methods for characterizing 2D material film quality, widely accepted test structures like TLM, Hall bar, capacitors, that need to be evaluated in detail. Why were they not employed? These are essential for understanding material quality issues, failure modes, contact resistance issues and so on. These details appear to be ignored in the approach.

3. Regarding this statement: "The generated results are reasonable since VT is primarily influenced by the top gate structure (metal work function and charge impurities/dipoles in the deposited gate dielectric), while μ depends on more factors, including the contact resistance and charge scattering." This is not correct because μ is an intrinsic property of the material. It is not related to contact resistance, if it is extracted properly. Apparently, the authors have used a two-point measurement to extract the mobility, and this calculated mobility indeed (falsely) depends on contact resistance. This should be clarified across the manuscript.

4. Figure 2f. First, grouping the data, as was done here, does not require machine learning. It is obvious to the naked eye what is a favourable outcome and what is not. Second, the figure leaves open the question about the other gray dots. Are they all from the same process flow? This would mean that the authors have in total only 5 process flows, which is not in agreement with the main text. The data points should be made distinguishable (if applicable).

5. The manuscript treats the device aspects as a black box, i.e. there is no correlation between measured values and the underlying physics. However, in device operation and optimization, this understanding is extremely important. It would be very interesting to understand how ML could be used to gain understanding in these intricate details?

6. Assuming that the main message of the paper is to establish Machine Learning for process optimization, I am wondering how the approach would work for a mature technology. In MoS₂, where there are extremely large differences, a simple algorithm can easily work. This has been shown in the past with statistical analysis for graphene and MoS₂ devices¹⁻⁴ It is easy, because the differences in devices and circuits are very large from process parameter to process parameter. It would be much more interesting to understand if the method can also work to optimize a mature technology, where the changes from run to run are miniscule.

7. In summary, the authors show some circuit examples and wafer scale fabrication. These represent in themselves nice results. However, these results are not discussed in detail and no insight is provided on the relationship between process, underlying physics phenomena and device / circuit performance. As such, this aspect of the paper does not contribute to the state of the art. Thus, the remaining aspect is the focus on machine learning. Here, the device / circuit results are not discussed in detail in respect to the machine learning process development, which arguably is the focus of the paper. Hence, the relevance of the ML procedures, also compared to previous

statistical analysis of wafer scale 2D materials¹⁻⁴ is not entirely clear.

References

1. Smith, A. D. et al. Wafer-Scale Statistical Analysis of Graphene FETs, Part I: Wafer-Scale Fabrication and Yield Analysis. *IEEE Transactions on Electron Devices* 64, 3919–3926 (2017).
2. Smith, A. D. et al. Wafer-Scale Statistical Analysis of Graphene Field-Effect Transistors, Part II: Analysis of Device Properties. *IEEE Transactions on Electron Devices* 64, 3927–3933 (2017).
3. Kim, Y. et al. Wafer-Scale Integration of Highly Uniform and Scalable MoS₂ Transistors. *ACS Appl. Mater. Interfaces* 9, 37146–37153 (2017).
4. Tian, M. et al. Wafer Scale Mapping and Statistical Analysis of Radio Frequency Characteristics in Highly Uniform CVD Graphene Transistors. *Advanced Electronic Materials* 5, 1800711 (2019).

Reviewer #3 (Remarks to the Author):

This manuscript presents optimized fabrication of CVD-grown MoS₂ for scalable circuits by involving the idea of machine learning. In this study, the MoS₂ is grown on sapphire with solid precursors of MoO₃ and S powders and the enhancement-mode FET are fabricated with gate-last process. The authors tend to conclude that device fabrication and performances are optimized by machine learning. Representative devices on digital, analog, and optoelectrical circuits are presented. Overall, this study is helpful for following research. The referee would further consider acceptance for publication if the authors could carefully address following issues:

1. In Fig. 2 a., the MoS₂ in the process is marked as 2-3nm (~3-5 layers), but the authors claim a monolayer MoS₂ film in the Fig. S1. Thickness of the grown sample, such as monolayer or few layer MoS₂ film, is significant to fabrication and performance. It is needed to confirm this issue and provide essential data, such as PL and Raman mapping of the representative devices. If the few layer MoS₂ is adopted for most demonstrations, please explain the reasons.
2. In Fig. 1.b, the seeding layer is partially deposited on the active region of the FET. Please explain why not fully deposited on the active region? The asymmetric FET design might cause some issues.
3. In fig. 2, the device is optimized with mobility with the V_{th} of ~2.1V. It would be ideal to include more discussion to explain how to realize the V_{th} tuning in loading transistor and keep the optimized mobility.
4. In this study, the authors mainly focus on mobility and V_{th} but more significant properties of the device are essential for real application, such as speed and power consumption.
5. Fabrication of top gate dielectrics on the surface of 2D materials is significant to device performances. It would be ideal to include more discussion on this issue and more details on ALD process of the high k dielectrics.
6. It seems that the devices are directly fabricated on sapphire wafer. Is it required to avoid damage in the transfer process for better electronic performances? It would be ideal to include more discussion on the issue because further fabrication or integration with the sapphire wafer might be issues.
7. All measurement of various logic circuit units are plotted in the time scale of seconds. It would be ideal to show high frequency output characteristics.
8. In fig. S17, photoresponse time of the device, such as raising and falling time, is in the scale of second. The performance might be a issue for real application. Is it due to interface issue or any possible reasons.

9. In fig. 3, the optimized performances are demonstrated with specific aspect ratio. This design of the device might raise issue on the speed or operations. Please explain this issue.

10. In most reported papers on the grown MoS₂, overall performances are usually determined with many issues, such as grain size, interface, crystallinity, defect density, variation in the batch synthesis and more process details. The issues might be highly coupled. It might be a bit difficult for readers to understand how machine learning could work for the optimization.

11. It is necessary to include detailed information on the fabricated devices, such as length/width and geometry of the FETs, material and size of the seeding layers, and thickness of the top dielectric.

Reviewer #4 (Remarks to the Author):

****THIS REPORT WAS WRITTEN IN COLLABORATION WITH REFEREE #2**

The manuscript "Wafer-Scale Functional Circuits Based on Two Dimensional Semiconductors with Fabrication Optimized by Machine Learning" by Xinyu Cheng and co-authors describes a Machine Learning based process optimization method. This this method was used to optimize the performance of MoS₂ channel field effect transistors. After the authors identified the optimized process, it was used to fabricate different circuits and to realize wafer scale MoS₂ device fabrication. While it is an interesting approach, the paper does not convincingly describe the advantage of the approach compared to conventional process development. In particular, it is not clear where the ML-based pattern recognition provides an advantage over classic design of experiment guidelines. I therefore think this manuscript is not appropriate for publication in Nature Communication.

Below are my detailed comments.

1. What is the advantage of the process optimization based on ML, over the traditional way (in which we optimize each step and then combine whatever we would like to have for a specific application). The traditional way of designing an experiment even appears to be more efficient, since one may not need to go through all fabrication process possibilities. In a clean DOE, one can run specific process combinations and just by looking at the data identify the most promising route. In the current form of the manuscript, this aspect is not discussed at all.

2. It is not clear for me what this statement indicates: "This is essential for materials, such as MoS₂ grown via CVD, which are synthesized on an insulating substrate, making device measurements after each processing step difficult." The authors should explain in more detail what they mean with this statement for the reader, as there are many established methods for characterizing 2D material film quality, widely accepted test structures like TLM, Hall bar, capacitors, that need to be evaluated in detail. Why were they not employed? These are essential for understanding material quality issues, failure modes, contact resistance issues and so on. These details appear to be ignored in the approach.

3. Regarding this statement: "The generated results are reasonable since VT is primarily influenced by the top gate structure (metal work function and charge impurities/dipoles in the deposited gate dielectric), while μ depends on more factors, including the contact resistance and charge scattering." This is not correct because μ is an intrinsic property of the material. It is not related to contact resistance, if it is extracted properly. Apparently, the authors have used a two-point measurement to extract the mobility, and this calculated mobility indeed (falsely) depends on contact resistance. This should be clarified across the manuscript.

4. Figure 2f. First, grouping the data, as was done here, does not require machine learning. It is obvious to the naked eye what is a favourable outcome and what is not. Second, the figure leaves open the question about the other gray dots. Are they all from the same process flow? This would mean that the authors have in total only 5 process flows, which is not in agreement with the main text. The data points should be made distinguishable (if applicable).

5. The manuscript treats the device aspects as a black box, i.e. there is no correlation between measured values and the underlying physics. However, in device operation and optimization, this understanding is extremely important. It would be very interesting to understand how ML could be used to gain understanding in these intricate details?

6. Assuming that the main message of the paper is to establish Machine Learning for process optimization, I am wondering how the approach would work for a mature technology. In MoS₂, where there are extremely large differences, a simple algorithm can easily work. This has been shown in the past with statistical analysis for graphene and MoS₂ devices^{1–4} It is easy, because the differences in devices and circuits are very large from process parameter to process parameter. It would be much more interesting to understand if the method can also work to optimize a mature technology, where the changes from run to run are miniscule.

7. In summary, the authors show some circuit examples and wafer scale fabrication. These represent in themselves nice results. However, these results are not discussed in detail and no insight is provided on the relationship between process, underlying physics phenomena and device / circuit performance. As such, this aspect of the paper does not contribute to the state of the art. Thus, the remaining aspect is the focus on machine learning. Here, the device / circuit results are not discussed in detail in respect to the machine learning process development, which arguably is the focus of the paper. Hence, the relevance of the ML procedures, also compared to previous statistical analysis of wafer scale 2D materials¹⁻⁴ is not entirely clear.

References

1. Smith, A. D. et al. Wafer-Scale Statistical Analysis of Graphene FETs, Part I: Wafer-Scale Fabrication and Yield Analysis. *IEEE Transactions on Electron Devices* 64, 3919–3926 (2017).
2. Smith, A. D. et al. Wafer-Scale Statistical Analysis of Graphene Field-Effect Transistors, Part II: Analysis of Device Properties. *IEEE Transactions on Electron Devices* 64, 3927–3933 (2017).
3. Kim, Y. et al. Wafer-Scale Integration of Highly Uniform and Scalable MoS₂ Transistors. *ACS Appl. Mater. Interfaces* 9, 37146–37153 (2017).
4. Tian, M. et al. Wafer Scale Mapping and Statistical Analysis of Radio Frequency Characteristics in Highly Uniform CVD Graphene Transistors. *Advanced Electronic Materials* 5, 1800711 (2019).

RESPONSE TO REVIEWERS

Reviewer# 1

The manuscript by Xinyu Chen et al. reports on the investigation of the key process parameters that impact the electrical characteristics of MoS₂ TG-TFTs by utilizing machine-learning (ML) algorithms. The authors fabricated enhancement-mode MoS₂ TG-FETs using ML-guided gate-last processing recipes and analysed them by 62-level SPICE modelling. They were able to demonstrate basic logic, analogue, and optoelectrical circuits. The results show that ML can be used for device optimization and shortening the learning cycle for novel 2D materials.

We would like to congratulate the authors on the nice results. The manuscript indicates the very well planned and nicely organised experiments. The paper is clearly structured and well written. The conclusions are supported by the obtained results. We would like to draw attention to only a few points.

Reply to the reviewer:

We sincerely appreciate the reviewer for carefully reading of the manuscript and positive comments on our work. It is very encouraging to receive your generous appraisal of our work, and we hope the worldwide scientific community can recognize our effort. We have modified the manuscript carefully and the detailed responses to your suggestions are listed below.

Q1: It is very common to perform transistor measurements in forward-and-back sweeping modes, which allows revealing the hysteresis. The latter can be a detrimental factor for the performance of the electrical circuits. It would be beneficial to discuss this matter and demonstrate the results by corresponding measurements.

Reply to the reviewer:

We thank the reviewer for the valuable suggestions. The hysteresis of transfer curves is indeed critical to the performance of 2D FETs and should be included in this work.

The hysteresis is originated from the complex interfaces, which have been widely accepted as the trapping/de-trapping processes of gate-oxide and oxide-channel interface [1]. These traps are mainly caused by the adsorbed impurities (water and gas molecules, etc.) on the channel surface [2] and the trapped charges in the channel, dielectric, and at the channel/dielectric interfaces [3-5]. The TG-FET structure adopted in our work excludes the influence of most impurity molecules, so the hysteresis of our device is mainly affected by the latter.

During the preparation of this reply, we measured the devices fabricated before to obtain the hysteresis characteristics, and the following has been added in the revised supplementary materials:

Table R1. Comparison group in which the SL material is a variable here and other parameters are kept the same. (Table S7 in SI)

Process	S/D	Seeding layer	Anneal of SL	Material of TG
a	Au	N/A	w/o	Au
b	Au	2 nm Y ₂ O ₃	w/o	Au
c	Au	2 nm Al ₂ O ₃	w/o	Au
d	Au	2 nm SiO ₂	w/o	Au

Fig. R1. The hysteresis characteristics of top-gated CVD MoS₂ FETs on a wafer-scale substrate with different processes (a-d in Table R1) at $V_{DS} = 0.1$ V. (Fig. S8 in SI)

Impact of seeding layer on the hysteresis characteristics for MoS₂ TG-FETs

In this group of comparison experiments, the hysteresis characteristics for MoS₂ TG-FETs prepared with various seeding layers (SLs) are investigated. The fabrication recipes and the

hysteresis characteristics for top-gated MoS₂ FETs are shown in Table S7 and Fig. S8. It is found that the process using 2nm SiO₂ as seeding layer has the smallest hysteresis, which indicates that the border trap density of the SiO₂/HfO₂ interface is less than that of the other conditions⁷. The hysteresis voltage is determined by the V_T difference between the dual-sweep transfer characteristic curves.

Meanwhile, the border trap density can be extracted by measuring the low-frequency noise, shown in Fig. R2 (Fig. S10 in SI). A relatively small border trap density of $3.5994 \times 10^{19} \text{ cm}^{-3} \text{ eV}^{-1}$ is estimated from the device with the smallest hysteresis (SiO₂/HfO₂ interface), lower than the value stated in previous reports [6-8].

In our previous work [9], we also systematically investigated the hysteresis of 2D FETs based on micro-scale mechanical exfoliated MoS₂ sheets by optimizing the TG material and structure, as shown in Fig. R3. Graphene TG gives a clean interface between TG and the dielectric layer, giving rise to a small FET hysteresis. Nevertheless, it was not adopted in this work because of its impracticality and complexity of wafer-scale fabrication.

Fig. R2. Low-frequency $1/f$ noise characteristics. (a) Normalized noise power spectra (S_{ID}/I_D^2) as a function of f . (b) S_{ID}/I_D^2 as a function of I_D at 100 Hz. The dashed line shows a corresponding linear fit.

Fig. R3. Dual-sweep I_D - V_{TG} curves of MoS₂/SiO₂/HfO₂ stack with (a) e-beam evaporated, (b) thermal evaporated, and (c) graphene TG.

Related references:

- [1] Guo, Y. et al. Charge trapping at the MoS₂-SiO₂ interface and its effects on the characteristics of MoS₂ metal-oxide-semiconductor field effect transistors. *Applied Physics Letters* **106**, 103109 (2015).
- [2] Yue, Q., Shao, Z., Chang, S. & Li, J. Adsorption of gas molecules on monolayer MoS₂ and effect of applied electric field. *Nanoscale Research Letters* **8**, 425 (2013).
- [3] Perera, M. M. et al. Improved Carrier Mobility in Few-Layer MoS₂ Field-Effect Transistors with Ionic-Liquid Gating. *ACS Nano* **7**, 4449-4458 (2013).
- [4] Park, Y., Baac, H. W., Heo, J. & Yoo, G. Thermally activated trap charges responsible for hysteresis in multilayer MoS₂ field-effect transistors. *Applied Physics Letters* **108**, 083102 (2016).
- [5] Kaushik, N. et al. Reversible hysteresis inversion in MoS₂ field effect transistors. *npj 2D*

Materials and Applications **1**, 34 (2017).

- [6] Srinivasan, P., Olubuyide, O., Choi, Y. S. & Marshall, A. in *International Electron Devices Meeting*. 27.24.21-27.24.24.
- [7] Sangwan, V. K. et al. Low-Frequency Electronic Noise in Single-Layer MoS₂ Transistors. *Nano Letters* **13**, 4351-4355 (2013).
- [8] Xie, X. et al. Low-Frequency Noise in Bilayer MoS₂ Transistor. *ACS Nano* **8**, 5633-5640 (2014).
- [9] Sheng, Y. et al. Gate Stack Engineering in MoS₂ Field-Effect Transistor for Reduced Channel Doping and Hysteresis Effect. *Advanced Electronic Materials* 2000395 (2020).

Q2: Throughout the report, the authors investigate a few key parameters for the performance of the transistors, such as mobility and threshold voltage. One of the important factors especially for extremely scaled transistors is contact resistance. Did the authors investigate the influence of the process parameters on the contact resistance?

Reply to the reviewer:

We thank the reviewer for the valuable suggestion. We strongly agree that the contact resistance (R_c) is a key parameter for the practical application of 2D FETs. A low R_c is critical to improving on-state current and high-frequency performance. Previous studies have shown that the high contact resistance stems from the unique interface between 2D semiconductors and 3D metals [10,11]. In modern Si CMOS technology, heavy doping through ion-implantation or alloying of the contact region is usually adopted to achieve an Ohmic-contact, but this is apparently not applicable to 2D materials because lattices can be easily destroyed due to their atomic thickness lattices [12]. Up to now, cutting-edge technologies have been developed to reduce R_c , such as chemical absorption doping [13], phase engineering [14] and tunneling contact [15], etc. Unfortunately, the advantages of these methods displayed in mechanically exfoliated samples are no longer evident for wafer-scale fabrication because of their uncontrollability, complexity, and incompatibility with industrial equipment.

To obtain a method suitable for large-scale preparation, we simply deposited four types of metals (Ti/Au, Au, In/Au and Ag/Au) as the source and drain electrodes, as shown in Fig. R4 (Fig. S2 in SI). The results of extracted R_c in Fig. R5 (Fig. S3 in SI) show that in this comparison group. Obviously, the devices with Au or Ti/Au electrodes exhibit a better contact and larger on-state current. We did not further examine the physical insight of these results because this work focuses on the optimization methodology of wafer-scale fabrication.

Fig. R4. I_D - V_{TG} curves of top-gated CVD MoS₂ FETs on a wafer-scale substrate with a, Ti/Au, b, Au, c, In/Au, and d, Ag/Au contacts at $V_{DS} = 0.1$ V. Blue and red curves correspond to the logarithmic scale on the left and linear scale on the right y-axis, respectively.

Fig. R5. Extracted contact resistance between MoS₂ and different metals using Y-function fitting to transfer curves.

Related references:

- [10] Das, S., Chen, H.-Y., Penumatcha, A. V. & Appenzeller, J. High Performance Multilayer MoS₂ Transistors with Scandium Contacts. *Nano Letters* **13**, 100-105 (2013).
- [11] Liu, H. et al. Statistical Study of Deep Submicron Dual-Gated Field-Effect Transistors on Monolayer Chemical Vapor Deposition Molybdenum Disulfide Films. *Nano Letters* **13**, 2640-2646 (2013).
- [12] Allain, A., Kang, J., Banerjee, K. & Kis, A. Electrical contacts to two-dimensional semiconductors. *Nature Materials* **14**, 1195-1205 (2015)
- [13] Fang, H. et al. Degenerate n-Doping of Few-Layer Transition Metal Dichalcogenides by Potassium. *Nano Letters* **13**, 1991-1995 (2013).
- [14] Lin, Y.-C., Dumcenco, D. O., Huang, Y.-S. & Suenaga, K. Atomic mechanism of the semiconducting-to-metallic phase transition in single-layered MoS₂. *Nature Nanotechnology* **9**, 391-396 (2014).
- [15] Wang, J. et al. High Mobility MoS₂ Transistor with Low Schottky Barrier Contact by Using Atomic Thick h-BN as a Tunneling Layer. *Advanced Materials* **28**, 8302-8308 (2016).

Q3: The results reveal a rather low frequency of the ring oscillator self-oscillations. Earlier results give values in the MHz region, which is two orders of magnitude larger than obtained in this work (<https://doi.org/10.1021/nl302015v>). The investigation of the low operating frequency may give insights into further improvement of the circuits.

Reply to the reviewer:

We thank the reviewer for the insightful suggestion.

The frequency is indeed a significant index for the ring oscillator (RO) device. The self-oscillation frequency is mainly influenced by the driving current, gate capacitance, parasitic capacitance, stage # and supply voltage, etc. We agree with the reviewer that its self-oscillation frequency is indeed lower than that obtained in the early work [16], which is mainly caused by the

following reasons: (1) In most of the previous results, mechanically exfoliated single crystalline MoS₂ sheets were adopted to fabricate RO devices, and the device performances of them are much higher than those wafer-scale polycrystalline MoS₂ films synthesized by the CVD method; (2) The channel length of the previously demonstrated devices (< 1 micron) is much smaller than the size in our work (tens of microns), which gives rise to a larger drive current, as well as a smaller gate capacitance; (3) In our MoS₂ transistors, a high parasitic capacitance further reduces the RO frequency due to a relatively large overlapped region between the gate and source/drain electrodes.

Considering the above reasons, it has a large room for future improvement of our MoS₂ circuits through optimization of fabrication techniques, such as fine alignment during successive lithography steps and further scaling down through Electron-beam lithography (In this work the laser direct-writer is used to perform lithography, which is fast but limited by a low resolution). However, at this moment, these are not the key issues of this work. To make this clear, we have modified the manuscript as follows:

The self-oscillation frequency of our RO s is relatively low compared with previous results¹¹, but there is a large room for future improvement via down-scale of device size.

Related references:

[16] Wang, H. et al. Integrated Circuits Based on Bilayer MoS₂ Transistors. *Nano Letters* **12**, 4674-4680 (2012).

Q4: Most of the measurements were performed in either DC mode or at a very low frequency. It seems important to discuss the possible operating frequencies and the limiting factors.

Reply to the reviewer:

We thank the reviewer for the constructive comment.

The following discussion has been added in the revised manuscript: *Besides measurement techniques, most previous literature reports about large-scale MoS₂ circuits also exhibit a low working frequency [17-19], and RC delay is the main limiting factor that restricts the operating frequency of this circuit. As mentioned above, the RC delay is mainly affected by the load capacitance and resistance, gate capacitance, parasitic capacitance, and equivalent*

resistance of the transistors in the circuit. To simplify the situation, we exclude the influence of additional load capacitance and resistance in the circuit, so the operating speed mainly depends on the response speed of the MoS₂ transistor, which is determined by the cut-off frequency $f_T = \frac{g_m}{2\pi C_G}$ [20], where g_m is the transconductance of the channel and C_G is the equivalent gate capacitance. In Fig. R6 (Fig. S15 added in the revised SI), we measured the experimental gate capacitance C_G , which is approximately 4.5 pF when $V_g \in [1.5, 3.0]$ V, and the transconductance $g_m \approx 3.8 \mu S$ when $V_g \in [1.5, 3.0]$ V. Therefore, the maximum value of f_T is approximately 134.5 kHz, which can be treated as a reference value of the possible circuit operating frequencies. This operating frequency can be further augmented by improving the MoS₂ material quality, further scaling down of transistor size, and better alignment during fabrication.

Fig. R6. Capacitance-voltage curves of the MoS₂ transistor at different sweep frequencies (1 kHz, 10 kHz, and 100 kHz). The insert is a measurement schematic diagram.

Related references:

- [17] Yu, L. et al. Design, Modeling, and Fabrication of Chemical Vapor Deposition Grown MoS₂ Circuits with E-Mode FETs for Large-Area Electronics. *Nano Letters* **16**, 6349-6356 (2016).
- [18] Li, N. et al. Large-scale flexible and transparent electronics based on monolayer molybdenum

disulfide field-effect transistors. *Nature Electronics* **3**, 711-717 (2020).

[19] Wang, L. et al. 2D Electronics: Electronic Devices and Circuits Based on Wafer-Scale Polycrystalline Monolayer MoS₂ by Chemical Vapor Deposition. *Advanced Electronic Materials* **5**, 1970038 (2019).

[20] Liu, Y. et al. Promises and prospects of two-dimensional transistors. *Nature* **591**, 43-53 (2021).

Q5: The optoelectrical applications were demonstrated by utilizing photodiodes. The authors demonstrate a high ON/OFF ratio. However, it is important to discuss the mode of operation (CW vs pulsed). It is well known that both these modes differ in such parameters as responsivity and ON/OFF ratio <https://doi.org/10.1021/nl502339q>

On the other hand, sensing applications require high-frequency operation and thus it is important to explicitly mention that the measurements were acquired in low-frequency mode.

Reply to the reviewer:

We thank the reviewer for suggesting this useful reference[21], in which the operating modes of the MoS₂ photodetectors were investigated systematically. We also cite this work in the revised manuscript.

For the photodetectors tested in this work, the dominant operation mode should be the photogating effect [21,22]. Under illumination, free electron-hole pairs are generated. The trap states are then occupied by photo-generated holes and act as a localized floating gate strongly modulating the channel conductance. Therefore, under illumination, the n-type transfer curves of MoS₂ FET are horizontally left-shifted compared with the dark state. Due to the slow detrapping process, the long lifetime of the photo-generated carriers results in high gain but relatively slow response speed, as shown in the supplementary Figure S20

Fig. S20. Time-resolved photoresponse of the TG MoS₂ phototransistor with $V_{TG} = -1$ V and $V_{DS} = 1$ V under illumination with $1.55 \text{ mW} \cdot \text{cm}^{-2}$ at 550 nm.

The response speed of the MoS₂ photodetector can be further improved by increase the crystalline quality of the MoS₂ or modifying the device structure. Thus, we adopted the comments from the reviewer and made the following modification in the supplementary information part 17:

Two mechanisms can influence the photoconductivity of a transistor: the photovoltaic (PV) and the photoconductivity (PC) effect²¹. The photovoltaic effect is described as a shift in transistor V_T due to charges transfer from the channel to the MoS₂/dielectric interface or nearby molecules, which makes the photodetectors respond slowly and show a considerable on-off ratio. The photoconductive effect is caused by the capture of carriers in band tail states in MoS₂ itself, and the response frequency is high while its response amplitude is relatively low. For the photodetector in this work, the dominant mode of operation should be the photogating effect^{21,22}. Under illumination, free electron-hole pairs are generated. The trap states are occupied by photo-generated holes and act as a localized floating gate strongly modulating the channel conductance. Therefore, the transfer curves under illumination are horizontally left shifted from that of the dark state. Due to the slow detrapping process, the long lifetime of the photo-generated carriers results in high gain but slow response speed. The response speed of MoS₂ photodetector can be further improved by modifying the structure and the operating mode.

Related references:

[21] Furchi, M. M., Polyushkin, D. K., Pospischil, A. & Mueller, T. Mechanisms of Photoconductivity in Atomically Thin MoS₂. *Nano Letters* **14**, 6165-6170 (2014).

[22] Fang, H. & Hu, W. Photogating in Low Dimensional Photodetectors. *Advanced Science* **4**, 1700323 (2017).

Q6: To make the comparison in table S7 more complete, it is worth to mention another recent work on IC on MoS₂ <https://www.nature.com/articles/s41928-020-0460-6>. Does the table state the average or maximum mobilities? I believe it makes sense to explicitly mention it.

Reply to the reviewer:

We thank the reviewer for the insightful suggestion. This is indeed a milestone literature by demonstrating fundamental building blocks of analog electronic devices [23]. We have added this paper in the revised manuscript and the revised Table S8 in the supplementary information. The mobility in the table has also been noted as the maximum mobility according to the suggestion.

Substrate Area	V_{DS} (V)	I_{on} (A)	I_{off} (A)	I_{on}/I_{off}	Max μ ($cm^2 V^{-1} s^{-1}$)	V_T (V)	Max.FET # in a working IC	W/L	Gate type	Ref
~50mm ²	5	9×10^{-5}	10^{-12}	10^8	~3	~0.65	115	45/2	BG	11
2mm × 3mm	2	2×10^{-5}	10^{-11}	10^6	3	~1.3	3	45/3	TG	12
4 inch	3	10^{-3}	10^{-13}	10^{10}	~55	~1.7	12	30/6	BG	13
-	3	10^{-3}	10^{-13}	10^{10}	~50	0.54	9	30/4	BG	14
1cm × 0.5cm	1	~ 10^{-5}	10^{-14}	10^8	> 40	-2	3	1/1	BG	15
-	1.5	10^{-5}	10^{-14}	10^9	80	2.41	10	30/4	BG	16
5mm × 5mm	8	12×10^{-5}	10^{-14}	10^{10}	~20	~3.2	12	4	BG	17
2 inch	0.5	5.65×10^{-5}	10^{-14}	10^9	~88	2.47	156	30/20	TG	This work

Table S8. Comparison of MoS₂ FET performance with recently published results

Related references:

[23] Polyushkin, D. K. et al. Analogue two-dimensional semiconductor electronics. *Nature Electronics* **3**, 486-491 (2020).

Q7: The yield of a 1-bit full-adder with 39 n-FETS is 50%. It is worth to discuss the reasons for the failures in non-working devices and possible routes to overcome these issues.

Reply to the reviewer:

We thank the reviewer for the constructive suggestion.

As we all know, there remain lots of obstacles for the wafer-scale application of 2D materials, including scalable synthesis and device processing [24-27]. In this work, the failed 1-bit full-adder circuits are all tested, and failure reasons were summarized as the gate leakage, high barrier contact, and unstable V_T drift, all of which can result in incorrect logical output in the adder circuit. The following explanation is added in the revised supplementary materials:

We attribute the low yield to two main issues: 1) The poor uniformity of MoS₂ film, such as grain boundaries and local defects, is detrimental to the yield of wafer-scale integrated circuits. It can be improved by a further upgrade of synthesis methods and tools. 2) The quality of our processing tools and cleanroom grade (class 10000) are relatively poor comparing with the industrial standard. A higher standard of facilities is necessary for future high-yield MoS₂ IC fabrication.

Related references:

[24] Kang, K. et al. High-mobility three-atom-thick semiconducting films with wafer-scale homogeneity. *Nature* **520**, 656-660 (2015).

[25] Yu, H. et al. Wafer-Scale Growth and Transfer of Highly-Oriented Monolayer MoS₂ Continuous Films. *ACS Nano* **11**, 12001-12007 (2017).

[26] Liu, Y. et al. Promises and prospects of two-dimensional transistors. *Nature* **591**, 43-53 (2021).

[27] Li, J. et al. Fractal-Theory-Based Control of the Shape and Quality of CVD-Grown 2D Materials. *Advanced Materials* **31**, 1902431 (2019).

Reviewer#2&4

The manuscript “Wafer-Scale Functional Circuits Based on Two Dimensional Semiconductors with Fabrication Optimized by Machine Learning“ by Xinyu Cheng and co-authors describes a Machine Learning based process optimization method. This this method was used to optimize the performance of MoS2 channel field effect transistors. After the authors identified the optimized process, it was used to fabricate different circuits and to realize wafer scale MoS2 device fabrication. While it is an interesting approach, the paper does not convincingly describe the advantage of the approach compared to conventional process development. In particular, it is not clear where the ML-based pattern recognition provides an advantage over classic design of experiment guidelines. I therefore think this manuscript is not appropriate for publication in Nature Communication.

Reply to the reviewer:

We sincerely thank the two reviewers for the careful reading and suggestive feedback, and we have made the following revision to our manuscript according to your valuable suggestions and comments.

Q1: What is the advantage of the process optimization based on ML, over the traditional way (in which we optimize each step and then combine whatever we would like to have for a specific application). The traditional way of designing an experiment even appears to be more efficient, since one may not need to go through all fabrication process possibilities. In a clean DOE, one can run specific process combinations and just by looking at the data identify the most promising route. In the current form of the manuscript, this aspect is not discussed at all.

Reply to the reviewer:

We thank the reviewer for the insightful suggestion. The advantage of the ML is indeed the core point of our work. Although we have tried to clarify it in our original manuscript, we agree that it was not clear enough and requires additional illustration. The following is a more detailed explanation: (sentences with *black italic* are added in the revised manuscript)

The most promising property of 2D semiconductors is the ultimate confinement in the perpendicular dimension, which is approximately several atoms thick. *Such intrinsic nature makes*

it extremely sensitive to the exterior environments and fabrication processing. For example, various processing recipes, exterior temperature, humidity, exposed atmosphere, etc., can influence the final device performance, especially for the top gate (TG) structure. In our research group, a detailed experimental record table, shown as below, has been used for years to fabricate TG MoS₂-FETs. Actually, all our data were collected by more than ten graduate students during the past five years.

Month/Day/Year:	Temperature (°C)	Humidity (%)			
Process step 1: material	a) Continuous monolayer CVD film; b) Monolayer + a small number of multi-layer points; c) Monolayer + a large number of multi-layer points; d) Continuous film				
S/D electrode	Process step 2: photores	a Vacuum baking	b Ordinary baking	Temperature of baking (°C)	Time of baking (min)
	Process step 3: exposure	rotate speed of LOR (r/s)	a The second layer (S181)	b The second layer (304)	Rotate speed of the second layer (r/s)
	Process step 4: develop	T1: Time of exposure (s)	dose		
	Process step 5: source/drain	T2: Time of development	a. underexposed, b. normal, c. overexposed		
	Process step 6: source/drain	a: Au	b: Ti/Au	c: In/Au	other
	Process step 7: Lift off	Deposition method	a: E-beam evaporation	b: Thermal evaporation	c: Sputter
	Process step 8: photores	Vacuum degree (Pa)	sticking layer (Å/s)	Au layer (Å/s)	Electron beam current (A)
Channel etching	Process step 9: photores	a: perfect	b: a few residues	c: difficult to lift off, requires ultra-sonic	Temperature (°C)
	Process step 10: etching process (Carrier gas, Time, Power, Vacuum)	a: with O ₂ clean before etching	b: without O ₂ clean before etching		
	Process step 11: Lift off	a: perfect	b: a few residues	c: difficult to lift off, requires ultra-sonic	Temperature (°C)
	Process step 12: annealing	a: mini annealing furnace	b: cvd annealing furnace	c: without annealing	
Dielectric deposition	Process step 13: seeding	a: high vacuum	b: annealing in Ar	c: Ar+H ₂	Temperature of annealing (°C)
	Process step 14: seeding layer annealing	a: Y	b: Al	c: 1nm SiO ₂ +1nm Al ₂ O ₃	d: 2nm SiO ₂
	Process step 15: High-K dielectric	e: 2nm SiO ₂ +2nm Al ₂ O ₃	f: no seeding layer		
Top electrode	Process step 16: metal sputtering	Temperature of annealing	Time of annealing (min)	a: O ₂	b: Ar+H ₂
	Process step 17: growth	The temperature of deposit	The thickness of deposit	a: with ALD cleaning	b: without ALD cleaning
	Process step 18: Lift off	a: Ti+Au	b: Au	c: Al	d: Pt
Hole etching	Process step 19: etching process (Carrier gas, Time, Power, Vacuum)	a: E-beam evaporation	b: Thermal evaporation	speed of evaporation (Å/s)	
	Process step 20: Lift off	a: perfect	b: a few residues	c: difficult to lift off, requires ultra-sonic	Temperature (°C)
Output of each step and device characteristics	a: CF ₄	b: O ₂	c: Ar	a: with O ₂ clean before etching	b: without O ₂ clean before etching
	hysteresis (V)	μ(cm ² /vs)	current ON/OFF ratio	Vth (V)	

Although we can optimize each step independently, and this is what we have been doing for the above list for years, and indeed we had accumulated lots of useful data. Nevertheless, we noticed that one could not simply “combine whatever we would like to have for a specific

application” for the TG FETs because successive processing steps are coupled together through the 2D channel interface and an ultrathin top dielectric layer, and thus can strongly influence each other, as shown below (Fig 1b in the manuscript).

Fig. R7. Schematic cross-section of an MoS₂ FET with TG and global BG. Various factors that influence the device performance are categorized.

For example, the contact fabrication that has already been optimized can still be influenced by the successive growth and annealing of the dielectric layer, as well as the top gate fabrication. *The most critical interface is the top surface 2D channel, although it is only directly contacted by the seeding layer, the ALD growth of high-k dielectric and TG fabrication can also influence the performance of the 2D channel, such as a drift of the V_T , because the thickness of TG dielectric is only tens of nanometers. Thus, all individual processing steps are highly coupled because any subsequent processing steps will influence the previous ones, making the processing optimization of 2D semiconductors more complicated than those in bulk semiconductors such as Si and Ge.*

Another example is that after the annealing of TG dielectric, not only the contact interface between the 2D semiconductor and the 3D metal electrodes is improved, it is also advantageous to repair the oxygen defects in the dielectric layer. Thus, the interface between the 2D channel and the dielectric layer can also be improved. We can list many more similar examples, and one can hardly verify one by one via careful characterizations and comparison experiments.

On the other hand, since the gate-last architecture (TG FET) is our goal, the traditional step-

by-step optimization is not practical, since it requires careful characterization after each step [28-30]. But in TG FETs, we can only measure the device after the TG is finalized (see more details in the next question). Therefore, if the traditional design of experiment (DOE) method is adopted, a large number of combinations are needed for comparison, which dramatically increases the research workload and reduces the optimization work efficiency.

As far as we know, the fabrication optimization methodology is also the key R&D topic for industrial IC fabrication foundries, which have already paid much attention to the so-called “yield ramp up”. Because in advanced technology nodes, such as the 7-nm-node which includes more than 3000 processing steps, it has become more challenging to do optimization and “search of key factors” only by human experience. Before we started this research project, **we collaborated with Samsung to optimize the fabrication processes through a machine learning algorithm, which indicates the recognition of such methodology by the industrial community.** However, we can not provide more details here due to our previous commercial agreement. Besides, **one of our corresponding authors (Jing Wan) has worked in Globalfoundries Inc. for years as a process integration engineer and, another co-author Ye Lu, has also worked in Intel Inc. for years as a process integration engineer (10-nm-node).** According to their experience, high- throughput data mining techniques have already been used extensively to analyze the data and optimize the process in the industry. **Prof. Lu also collaborated with a famous semiconductor consulting company “PDF solution Inc.”, which also uses similar machine learning algorithms (“expert system”) to provide their service for foundries, as shown below:**

The following has been added to the revised manuscript:

The synthesis of wafer-scale MoS₂ and other 2D semiconductors is currently under fast development [31-32], providing more candidates for fabricating FETs and ICs. Even for the MoS₂ film itself investigated in this work, the synthesis method can be further optimized to modify the grain size, crystallinity, defect density, etc., which all influence the overall performances of the MoS₂ FETs. This is one of the main reasons why academic researchers have opted not to undertake strenuous efforts on the fabrication optimization of specific 2D semiconductors. Therefore, our results can be extended to other 2D semiconductors and emerging novel materials to reduce their device optimization burdens and shorten the learning cycle.

Related references:

- [28] Baugher, B. W. H., Churchill, H. O. H., Yang, Y. & Jarillo-Herrero, P. Intrinsic Electronic Transport Properties of High-Quality Monolayer and Bilayer MoS₂. *Nano Letters* **13**, 4212-4216 (2013).
- [29] Mori, T. et al. in *2015 IEEE 15th International Conference on Nanotechnology (IEEE-NANO)*. 762-765.
- [30] Sheng, Y. et al. Gate Stack Engineering in MoS₂ Field-Effect Transistor for Reduced Channel Doping and Hysteresis Effect. *Advanced Electronic Materials* 2000395 (2020).
- [31] Butler, K. T., Davies, D. W., Cartwright, H., Isayev, O. & Walsh, A. Machine learning for molecular and materials science. *Nature* **559**, 547-555 (2018).
- [32] Raccuglia, P. et al. Machine-learning-assisted materials discovery using failed experiments. *Nature* **533**, 73-76 (2016).

Q2: It is not clear for me what this statement indicates: “This is essential for materials, such as MoS₂ grown via CVD, which are synthesized on an insulating substrate, making device measurements after each processing step difficult.” The authors should explain in more detail what they mean with this statement for the reader, as there are many established methods for characterizing 2D material film quality, widely accepted test structures like TLM, Hall bar, capacitors, that need to be evaluated in detail. Why were they not employed? These are essential for understanding material quality issues, failure modes, contact resistance issues and so on. These details appear to be ignored in the approach.

Reply to the reviewer:

We thank the reviewer for the constructive suggestion. We apologize for the unclear description in the previous version, and the following is a detailed explanation:

First of all, mechanically exfoliated MoS₂ sheets are usually transferred to a Si wafer covered by a SiO₂ dielectric layer, and back gated (BG) transfer curves (I_d - V_{bg}) can be conveniently characterized for such MoS₂ BG-FETs, which have been extensively investigated for the past few years. However, for wafer-scale MoS₂ continuous film, the best synthesis method is known as the CVD growth on an insulating sapphire substrate. Therefore, it is only suitable for the top gate (TG) transistor structure (i.e., it requires gate-last fabrication). On the other hand, TG architecture is favored by practical device application and circuit-level integration because of its independent gate control for practical device operation. Nevertheless, it is rather difficult to form a uniform and high-quality dielectric layer on 2D semiconductors as they lack dangling bonds on the surface, and most of the reported MoS₂-based TG FETs with an ALD high- k dielectric layer suffer from a severe n-doping effect due to the sulfur vacancies. To overcome this problem, adding a seeding layer (SL) is a widely accepted solution for interface engineering, but it also brings extra interfacial charge traps or dipoles, which makes the MoS₂ interface more complex.

Therefore, it is challenging to perform a step-by-step optimization for a MoS₂ TG structured device. Because, *without a BG, the transfer characteristics (I_d - V_d output curve is doable, but not much information can be extracted) can only be conducted after the final step (TG metalization). So, all influencing factors from multiple processing steps are highly coupled to complicate the understanding of each processing step.*

Regarding test structures like TLM, Hall bar, capacitor method, they are indeed helpful for investigating fundamental device physics. But, they can not provide direct information for the FET performance (such as field-effect mobility, V_T , SS , ON and OFF current, etc.). In fact, the TLM, 4-probe, and capacitor measurements are all included in our previously reported results about wafer-scale MoS₂ devices [30,33], but they are not the focus of this work.

Related references:

[33] Xu, H. et al. High-Performance Wafer-Scale MoS₂ Transistors toward Practical Application.

Small **14**, 1803465 (2018).

Q3: Regarding this statement: “The generated results are reasonable since V_T is primarily influenced by the top gate structure (metal work function and charge impurities/dipoles in the deposited gate dielectric), while μ depends on more factors, including the contact resistance and charge scattering.” This is not correct because μ is an intrinsic property of the material. It is not related to contact resistance, if it is extracted properly. Apparently, the authors have used a two-point measurement to extract the mobility, and this calculated mobility indeed (falsely) depends on contact resistance. This should be clarified across the manuscript.

Reply to the reviewer:

We thank the reviewer for the insightful suggestion. We agree that our previous illustration was inaccurate and a little misleading. Here we provide a more explicit discussion:

As mentioned by the referee, intrinsic mobility is only related to the properties of the material and not related to the contact. The theoretical calculation estimated an intrinsic mobility value of $\sim 410 \text{ cm}^2/\text{Vs}$ at room temperature, based on the first-principle calculation of electron-phonon interaction [35,36]. However, as far as we know, no reported MoS₂ FETs can achieve such high mobility, and most published papers apply the following equation to estimate a field-effect mobility:

$$\mu_{gm} = \frac{dI_{DS}}{dV_{TG}} \cdot \frac{L}{W} \cdot \frac{1}{C_{OX}V_{DS}}$$

where $\frac{L}{W}$ is the length/width ratio of the channel, C_{OX} is the gate oxide capacitance, and $\frac{dI_{DS}}{dV_{TG}}$ is the slope of the liner regime of the transfer curve. Such calculated mobility is apparently influenced by the contact resistance [34], which is mainly induced by the Schottky barrier between the 2D semiconductor and the 3D metal electrodes.

The mobility extraction methodology adopted in our work is the so-called “Y-function method”, which has been widely used Si FETs and also suitable for 2D FETs [37-38]:

Y-function is defined as $Y = \frac{I_{DS}}{\sqrt{g_m}}$, where I_{DS} is the drain current and g_m is the transconductance ($=\frac{dI_{DS}}{dV_{TG}}$). Mobility can be estimated from Y-function by the equation $\mu_Y = \left(\frac{Y}{V_{TG}-V_T}\right)^2 \left(\frac{L}{WC_{OX}V_{DS}}\right)$, where $\frac{L}{W}$ is the length/width ratio of the channel, C_{OX} is the gate oxide capacitance and V_T is estimated by linear extrapolation from a plot of Y vs. V_{GS} rather than I_{DS} vs. V_{GS} .

Through the Y-function, the extracted mobility can theoretically rule out the impact of contact resistance. Although, as discussed in the recently published paper [34], μ_Y still depends on the contact resistance R_c , it is more accurate than that of μ_{gm} . The “4-probe method” is indeed more accurate since the contact resistance can be excluded entirely. However, the “Y-function” method is more convenient since it is performed directly on the fabricated MOSFET, and thus widely used for 2D FETs [37-38].

In order to avoid misunderstanding, we have modified the manuscript:

“The generated results are reasonable upon physical analysis, since V_T is primarily influenced by the TG structure (metal work function and charge impurities/dipoles in the deposited gate dielectric). At the same time, the mobility μ is extracted by the Y-function method, which depends on multiple factors such as interfacial scattering and contact resistance⁴².”

Related references:

- [34] Sebastian, A., Pendurthi, R., Choudhury, T. H., Redwing, J. M. & Das, S. Benchmarking monolayer MoS₂ and WS₂ field-effect transistors. *Nature Communications* **12**, 693 (2021).
- [35] Gunst, T., Markussen, T., Stokbro, K. & Brandbyge, M. First-principles method for electron-phonon coupling and electron mobility: Applications to two-dimensional materials. *Physical Review B* **93**, 035414 (2016).
- [36] Li, X. et al. Intrinsic electrical transport properties of monolayer silicene and MoS₂ from first principles. *Physical Review B* **87**, 115418 (2013).
- [37] Zheng, X. et al. Patterning metal contacts on monolayer MoS₂ with vanishing Schottky

barriers using thermal nanolithography. *Nature Electronics* 2, 17-25 (2019).

[38] Smithe, K. K. H., Suryavanshi, S. V., Muñoz Rojo, M., Tedjarati, A. D. & Pop, E. Low Variability in Synthetic Monolayer MoS₂ Devices. *ACS Nano* 11, 8456-8463 (2017).

Q4: Figure 2f. First, grouping the data, as was done here, does not require machine learning. It is obvious to the naked eye what is a favourable outcome and what is not. Second, the figure leaves open the question about the other gray dots. Are they all from the same process flow? This would mean that the authors have in total only 5 process flows, which is not in agreement with the main text. The data points should be made distinguishable (if applicable).

Reply to the reviewer:

We thank the reviewer for the constructive suggestion. We also apologize for the unclear explanation of the data in Fig. 2f.

To clarify this, the original gray dots are replaced by the colored dots in the revised Fig. 2f. Each color corresponds to one processing combination. Most processing combinations were designed by our experiences based on step-by-step optimization. The main text is revised as:

“We then demonstrate that ML can also be used to co-optimize all process parameters, as shown in Fig. 2d. After the EL training, a score predictor can predict the results from a specific processing combination (i.e., one process recipe). All possible process recipes are then sorted using a grid search method, as shown in Fig. 2e. To demonstrate this, we fabricated more than 500 MoS₂ FETs, which are summarized in the μ - V_T plot in Fig. 2f. Each color corresponds to FETs fabricated by one process recipe. Most recipes were designed by human experiences based on step-by-step optimization. For example, one recipe provides a high μ value (orange circles), and another recipe provides a positive V_T value (blue circles). However, the mixing of two recipes (green circles) can not guarantee both high μ and positive V_T , mainly due to crosstalk between different processing steps. Therefore, the combination of multiple steps with each optimized does not necessarily generate the best device. We then fabricate a batch of devices (red stars in Fig. 2f) following the suggestion of the sorting result (red arrow in Fig. 2e). This recipe combination

(details see Supplementary Table 6) also gives rise to an average μ about $75 \text{ cm}^2/\text{V}\cdot\text{s}$ and V_T of 2.1 V , as well as a high wafer-scale uniformity that is important for large-scale circuits, as shown in Fig. 2g.”

Fig. R8 (Fig. 2f). More than 500 MoS₂ TG-FETs summarized in a μ - V_T plot. Each color corresponds to one process recipe. The red stars are the results of the process recipe in e pointed by the red arrow.

Q5: The manuscript treats the device aspects as a black box, i.e. there is no correlation between measured values and the underlying physics. However, in device operation and optimization, this understanding is extremely important. It would be very interesting to understand how ML could be used to gain understanding in these intricate details?

Reply to the reviewer:

We thank the reviewer for the kind suggestion. We strongly agree with the referee that the ML method is quite like a black box, which is the working principle of ML (simply speaking, a statistical classification algorithm for large data sets), and people have already applied ML methods to assist the searching of low-dimensional materials [31,32]. *Although ML represents a computer-aided learning process from large amounts of device data, the optimization process*

guided by ML also reveals some underlying physics that can explain our experimental data.

For example, in Table R2 (Table S6 in the supplementary information), recipe *c* has the highest score ranked by the ML algorithm, in which both contact (Ti/Au) and seeding layer (SiO₂) are not the best options obtained by single-step optimizations. It can be partially explained by: 1) the addition of Ti as a buffer layer between MoS₂ film and Au electrodes can effectively increase the adhesion of contacts, and its slightly smaller work function is beneficial to reduce the contact resistance and increase the on-state current [39]. 2) A 2-nm-thick SiO₂ can reduce the damage to MoS₂ by the growth of HfO₂, but its defect level is also high because of the physical vapor deposition method. However, the subsequent annealing can likely repair the oxygen defects in the SiO₂ layer, thus substantially improving the quality of the dielectric layer and the transistor's electrostatic control capability [30].

Table R2. Comparison group with 3 different recipe combinations.

Process combination	S/D	Seed layer	Anneal of SL	Material of TG
a	Au	2 nm Al ₂ O ₃	W/	Au
b	Au	2 nm SiO ₂	W/	Au
c	Ti/Au	2 nm SiO ₂	W/	Au

Although understanding the underlying physics is not the key to this work, we believe our work opens the door for future detailed investigation of applying ML to optimize device fabrication. It also leaves open questions for researchers interested in the device physics of 2D semiconductors.

Related references:

[39] Liu, H. et al. Switching Mechanism in Single-Layer Molybdenum Disulfide Transistors: An Insight into Current Flow across Schottky Barriers. *ACS Nano* **8**, 1031-1038 (2014).

Q6: Assuming that the main message of the paper is to establish Machine Learning for process

optimization, I am wondering how the approach would work for a mature technology. In MoS2, where there are extremely large differences, a simple algorithm can easily work. This has been shown in the past with statistical analysis for graphene and MoS2 devices¹⁻⁴ It is easy, because the differences in devices and circuits are very large from process parameter to process parameter. It would be much more interesting to understand if the method can also work to optimize a mature technology, where the changes from run to run are miniscule.

Reply to the reviewer:

We thank the reviewer for the insightful suggestion.

Since machine learning is a data analytics technique that teaches computers to learn from accumulated experimental data, this method can be adopted in most areas as long as the data set is sufficient. In fact, the ML optimization method is even more suitable for mature technologies since the stable process can generate a large amount of data with less variability and noise.

As mentioned in the answer to Q1, **one of our corresponding authors (Jing Wan) has worked in Globalfoundries Inc. for years as a process integration engineer and, another co-author Ye Lu, has also worked in Intel Inc. for years as a process integration engineer (10-nm-nodel).** According to their experience, data mining techniques have already been used extensively to analyze the data and optimize the process in the industry.

Q7: In summary, the authors show some circuit examples and wafer scale fabrication. These represent in themselves nice results. However, these results are not discussed in detail and no insight is provided on the relationship between process, underlying physics phenomena and device / circuit performance. As such, this aspect of the paper does not contribute to the state of the art. Thus, the remaining aspect is the focus on machine learning. Here, the device / circuit results are not discussed in detail in respect to the machine learning process development, which arguably is the focus of the paper. Hence, the relevance of the ML procedures, also compared to previous statistical analysis of wafer scale 2D materials¹⁻⁴ is not entirely clear.

References

1. Smith, A. D. et al. Wafer-Scale Statistical Analysis of Graphene FETs, Part I: Wafer-Scale Fabrication and Yield Analysis. IEEE Transactions on Electron Devices 64, 3919–3926 (2017).

2. Smith, A. D. et al. Wafer-Scale Statistical Analysis of Graphene Field-Effect Transistors, Part II: Analysis of Device Properties. IEEE Transactions on Electron Devices 64, 3927–3933 (2017).

3. Kim, Y. et al. Wafer-Scale Integration of Highly Uniform and Scalable MoS₂ Transistors. ACS Appl. Mater. Interfaces 9, 37146–37153 (2017).

4. Tian, M. et al. Wafer Scale Mapping and Statistical Analysis of Radio Frequency Characteristics in Highly Uniform CVD Graphene Transistors. Advanced Electronic Materials 5, 1800711 (2019).

Reply to the reviewer:

Again, we sincerely thank the reviewer #2 & 4 for their constructive suggestions. We hope the above reply can address the questions raised by them.

To summarize, the focus of this work is to optimize the fabrication process efficiently for the appropriate device performance in line with the requirements of MoS₂ circuits. We do not focus on one parameter, such as mobility and subthreshold swing, and we also agree that the performances of our devices are not “state of the art”. However, unlike other research [40-43] that focuses on optimizing one individual factor, various factors in many variables and process combinations are comprehensively considered with the assistance of a machine learning algorithm. Therefore, we focus on a more comprehensive optimization of the MoS₂ FETs for wafer scale application, which is pretty challenging, as already explained above. As far as we know, no results show working TG structured enhance-mode MoS₂ FETs under a satisfactory wafer-scale uniformity.

Regarding the “*the device/circuit results are not discussed in detail in respect to the machine learning process development*”, it has been added in Supplementary Fig. S9 and discussed in section 18, where we can see that both processing and material uniformity are critical to the yield of circuits:

We attribute that the relatively low wafer-scale yield to three main issues:

1) The quality and uniformity of MoS₂ films, such as grain boundaries and local defects, is detrimental to the yield of wafer-scale integrated circuits. It can be improved by a further upgrade of synthesis methods and facilities.

2) As shown in Fig. S9, the uniformity of MoS₂ FETs still depends on the processing recipes. The transistor size might also influence the yield since the length and width of the MoS₂ FETs are not variable parameters in our ML algorithm.

3) The quality of processing tools and cleanroom grade (10⁴) is relatively low comparing with the industrial standard. A higher standard laboratory is necessary for future serious MoS₂ IC fabrication.

The suggested references are indeed important works about wafer-scale 2D materials. We have already cited them in the revised manuscript. The devices in reference 3 are back gate structured, while graphene devices in ref. 1, 2, and 4 are relatively simple to fabricate.

Finally, we thank the referee again and hope that the previous answers have resolved most of the reviewers' concerns.

Related references:

[40] Jung, Y. et al. Transferred via contacts as a platform for ideal two-dimensional transistors. *Nature Electronics* **2**, 187-194 (2019).

[41] Wang, Y. et al. Van der Waals contacts between three-dimensional metals and two-dimensional semiconductors. *Nature* **568**, 70-74 (2019).

[42] Farmer, D. B. et al. Utilization of a Buffered Dielectric to Achieve High Field-Effect Carrier Mobility in Graphene Transistors. *Nano Letters* **9**, 4474-4478 (2009).

[43] Li, W. et al. Uniform and ultrathin high- κ gate dielectrics for two-dimensional electronic devices. *Nature Electronics* **2**, 563-571 (2019).

Reviewer#3

This manuscript presents optimized fabrication of CVD-grown MoS₂ for scalable circuits by involving the idea of machine learning. In this study, the MoS₂ is grown on sapphire with solid precursors of MoO₃ and S powders and the enhancement-mode FET are fabricated with gate-last

process. The authors tend to conclude that device fabrication and performances are optimized by machine learning. Representative devices on digital, analog, and optoelectrical circuits are presented. Overall, this study is helpful for following research. The referee would further consider acceptance for publication if the authors could carefully address following issues:

Reply to the reviewer:

We thank the reviewer for carefully reviewing our manuscript and raising many insightful comments. We also appreciate the reviewer for recognizing that our study “*is helpful for following research*” along with a positive recommendation.

Q1: In Fig. 2 a., the MoS₂ in the process is marked as 2-3nm (~3-5 layers), but the authors claim a monolayer MoS₂ film in the Fig. S1. Thickness of the grown sample, such as monolayer or few layer MoS₂ film, is significant to fabrication and performance. It is needed to confirm this issue and provide essential data, such as PL and Raman mapping of the representative devices. If the few layer MoS₂ is adopted for most demonstrations, please explain the reasons.

Reply to the reviewer:

We greatly appreciate the reviewer for pointing out this problem in Fig. 2a. The MoS₂ thickness noted as “2-3nm (~3-5 layers)” was indeed a mistake made by us, which should be corrected as 0.8nm (~ monolayer).

To confirm this, Raman and PL spectra are characterized for our MoS₂ film, shown in Fig. R9(a-b). An additional AFM image is also shown in Fig. R9(c) also indicates that the thickness is about ~0.8nm. All these characterization results are consistent with those of monolayer MoS₂ previously reported [44,45]. The Raman mapping over the whole wafer is also shown in Fig. 1a in the main text in order to show a satisfying uniformity of our wafer-scale MoS₂ film. Therefore, we modified the error in Fig. 2a in the main text, shown in Fig. R10.

Fig. R9. (a) Raman spectrum and (b) PL spectrum of monolayer MoS₂ at the channel of 2D MoS₂ FETs. (c) AFM image of monolayer MoS₂ film. The scratch is intentionally made to reveal the edge. The insert is the height profile along with the white dashed line in the AFM image.

Fig. R10. Process flow for fabricating TG MoS₂ FETs. The variations in each step are marked in blue.

Related references:

[44] Splendiani, A. et al. Emerging Photoluminescence in Monolayer MoS₂. *Nano Letters* **10**, 1271-1275 (2010).

[45] Li, H. et al. From Bulk to Monolayer MoS₂: Evolution of Raman Scattering. *Advanced Functional Materials* **22**, 1385-1390 (2012).

Q2: In Fig. 1.b, the seeding layer is partially deposited on the active region of the FET. Please explain why not fully deposited on the active region? The asymmetric FET design might cause some issues.

Reply to the reviewer:

We thank the reviewer for the careful reading.

During the practical processing of preparing the device, the seeding layer actually fully covers the entire channel region for the following growth of the high- κ dielectric layer, which means the device structure is symmetric as the conventional transistors. Fig. 1b is merely a schematic view, and it can more clearly show the cross-sectional view of the gate stack with several different interfaces, and the seeding layer acts as a buffer layer between the MoS₂ channel and the high- κ dielectric layer (HfO₂).

In the revised manuscript, we add an explanation in the Figure caption: “... *the seeding layer is actually fully deposited on the complete channel region....*”

Q3: In fig. 2, the device is optimized with mobility with the V_{th} of $\sim 2.1V$. It would be ideal to include more discussion to explain how to realize the V_{th} tuning in loading transistor and keep the optimized mobility.

Reply to the reviewer:

We thank the reviewer for the insightful suggestion.

Multiple factors can influence the V_T of the FETs, including 1) material type, thickness and deposition method of seeding layer; 2) material type, thickness and growth temperature of high- k material; 3) material type of the top gate electrode. An overall optimization is extremely difficult. This is also the motivation for applying the machine learning method.

In the revised supplementary material, one more section is added for illustrating the influence of gate metal:

Based on the above results, we can tell that the gate metals with different work functions can control the V_T of the FETs with little effect on the mobility, and the band diagrams of FETs with gate metal work function either greater or smaller than the MoS₂ work function are shown in Fig.

S7[46]. At zero gate bias, the gate metal with a low work function tends to attract more electrons in the channel, tuning the channel to the charge accumulation regime, while a high work function metal does the opposite. For the Al- and Au-gated MOS₂ FETs, the V_T shift is about 2 V and shifts its V_T from negative to positive. Thus it is beneficial to develop a direct-coupled FET logic (DCFL) [46,47]. Moreover, in this V_T engineering method, mobility is not influenced because the interface quality between the channel and the dielectric layer is maintained, i.e., no additional carrier scattering is introduced.

Fig. R11. Energy band diagrams for (a) isolated metal, insulator and semiconductor, and (b) after their intimate contact and thermal equilibrium is established [46].

Related references:

[46] Wang, H. et al. Integrated Circuits Based on Bilayer MoS₂ Transistors. *Nano Letters* **12**, 4674-4680 (2012).

[47] Ayers, J. E. Digital integrated circuits: analysis and design; CRC Press: Boca Raton, FL, 2004.

Q4: In this study, the authors mainly focus on mobility and V_{th} but more significant properties of the device are essential for real application, such as speed and power consumption.

Reply to the reviewer:

We thank the reviewer for the constructive comment.

Actually, one of the determinants of the speed is the RC delay which is mainly affected by the gate capacitance, parasitic capacitance, and equivalent resistance of the transistors. It is analyzed in detail in the following questions.

Regard the device speed, RC delay is the main limiting factor that restricts the operating frequency of this circuit. The RC delay is mainly affected by the load capacitance and resistance, gate capacitance, parasitic capacitance, and equivalent resistance of the transistors in the circuit. To simplify the situation, we exclude the influence of additional load capacitance and resistance in the circuit, so the operating speed is mainly determined by the cut-off frequency $f_T = \frac{g_m}{2\pi C_G}$ of the MoS₂ transistor, where g_m is the transconductance of the channel and C_G is the equivalent gate capacitance. In Fig. R12 (*Fig. S15 added in the revised SI*), we measured the experimental gate capacitance C_G , which is approximately 4.5 pF when $V_g \in [1.5, 3.0]$ V, and the transconductance $g_m \approx 3.8 \mu\text{S}$ when $V_g \in [1.5, 3.0]$ V. Therefore, the maximum value of f_T is approximately 134.5 kHz, which can be regarded as a reference value of the possible circuit operating frequencies.

Fig. R12. Capacitance-voltage curves of the MoS₂ transistor at different sweep frequencies (1 kHz, 10 kHz, and 100 kHz), the insert is a measurement schematic diagram.

The power consumption of an FET is mainly divided into static, dynamic, and short circuit power consumptions [48]. Here is a brief discussion:

- 1) The static power consumption is caused by the static conductive path or leakage current between the power supply and the ground. It can be calculated by $P_{stat} = I_{stat}V_{DD}$, where I_{stat} is the static path current, V_{DD} is the supply voltage. The static power consumption can be improved by reducing the static path current, the leakage current I_{leak} , and V_{DD} . For MoS₂ FETs, the current On/Off ratio is the key.
- 2) Dynamic power consumption occurs only at the moment the FET switches and is mainly caused by the charge and discharge processes of the load capacitors. It is calculated by the formula $P_{dyn} = C_L V_{DD}^2 f_{0 \rightarrow 1}$, where C_L is the load capacitor, V_{DD} is the supply voltage, $f_{0 \rightarrow 1}$ is the flip frequency that consumes energy. The dynamic power consumption can be lowered by reducing the load capacitance, the operating voltage, and the operating frequency.
- 3) Short-circuit power consumption, like dynamic power consumption, occurs only at the moment when the FET switches and is mainly caused by the temporary current path between the power supply and the ground. It is calculated by the formula $P_{dp} = t_{sc} V_{DD} I_{peak} f = C_{sc} V_{DD}^2 f$, where t_{sc} is the time for pull-up network and pull-down network operating simultaneously, V_{DD} is the supply voltage, I_{peak} is the peak current, C_{sc} is the capacitance when pull-up network and pull-down network are simultaneously conducting. The short-circuit power consumption can be reduced by reducing the on-state current, the operating voltage, and the operating frequency.

In general, it is necessary to manipulate all of the FET parameters, including load capacitance, the on-state current, V_{DD} , V_T , I_{on}/I_{off} , SS , and the operating frequency to optimize the overall speed and power consumption, as shown in Fig. R13 (Fig. 1c in the main text).

Fig. R13. Schematic diagram of the relationship between performance parameters of the transistor and performance limitations of the integrated circuit.

Related references:

[48] Rabaey, J. M., Chandrakasan, A. P., B Nikolić. Digital integrated circuits: a design perspective[M]. Prentice Hall, 2003.

Q5: Fabrication of top gate dielectrics on the surface of 2D materials is significant to device performances. It would be ideal to include more discussion on this issue and more details on ALD process of the high k dielectrics.

Reply to the reviewer:

We thank the reviewer for the beneficial suggestion.

The deposition methods of large-scale uniform high- κ insulating materials via ALD are also critical in Si CMOS devices and have been developed for years. Those methods can also be adopted in 2D FETs [49-51]. ALD is the self-limiting growth mode in which saturation can be achieved at each step of the reaction process under appropriate conditions. However, due to the absence of dangling bonds on the surface of 2D materials, the direct growth of high- κ dielectric layer such as HfO₂ is rather tricky. Therefore, the seeding layers such as SiO₂, Al₂O₃ and YO₂ are commonly adopted as a buffer layer between the high- κ dielectric and the channel to ensure the quality of the high- κ dielectric layer and perfect device performance [52,53].

To include more details on the ALD process of the high k dielectrics, we also added the following detailed recipe description in the revised Supplementary Materials:

“... A SiO₂ seeding layer is deposited by E-beam evaporation and subsequently annealed in an oxygen atmosphere at 200 °C. Then HfO₂ layer is grown by atomic layer deposition (ALD) at 180 °C, where the Tetrakis(dimethylamino)hafnium [Hf(N(CH₃)₂)₄] and H₂O are used as the precursors and nitrogen (N₂) as the carrier gas. The sequence of pulses for one deposition cycle of HfO₂ is H₂O (0.01 s)/N₂ (120 s)/ Hf(N(CH₃)₂)₄ (0.1 s)/N₂ (100 s). Another group of lithography/lift-off/deposition processes is repeated to form the top metal layer. For electrical probing or further fabrication of more complex circuits, SF₆ plasma etching is used to remove the HfO₂ layer on top of the source/drain electrodes to form contact via.”

Related references:

- [49] McDonnell, S. et al. HfO₂ on MoS₂ by Atomic Layer Deposition: Adsorption Mechanisms and Thickness Scalability. *ACS Nano* **7**, 10354-10361 (2013).
- [50] Radisavljevic, B., Radenovic, A., Brivio, J., Giacometti, V. & Kis, A. Single-layer MoS₂ transistors. *Nat Nanotechnol* **6**, 147-150 (2011).
- [51] Wang, L. et al. 2D Electronics: Electronic Devices and Circuits Based on Wafer-Scale Polycrystalline Monolayer MoS₂ by Chemical Vapor Deposition. *Advanced Electronic Materials* **5**, 1970038 (2019).
- [52] Zou, X. et al. Interface Engineering for High-Performance Top-Gated MoS₂ Field-Effect Transistors. *Advanced Materials* **26**, 6255-6261 (2014).
- [53] Sheng, Y. et al. Gate Stack Engineering in MoS₂ Field-Effect Transistor for Reduced Channel Doping and Hysteresis Effect. *Advanced Electronic Materials* 2000395 (2020).

Q6: It seems that the devices are directly fabricated on sapphire wafer. Is it required to avoid damage in the transfer process for better electronic performances? It would be ideal to include

more discussion on the issue because further fabrication or integration with the sapphire wafer might be issues.

Reply to the reviewer:

We thank the reviewer for the kind suggestion.

Indeed, the high-quality MoS₂ films are synthesized on the sapphire wafer (111 surface). The top gated MOSFETs and circuits are directly fabricated with the wafer-scale MoS₂ film on the sapphire wafer without using transfer in order to avoid degradation of film quality.

The sapphire wafer only acts as a rigid insulating substrate to support the MoS₂ integrated circuits. This structure is similar to the conventional silicon on insulators (SOI) [54,55] technology already used in the CMOS industry. SOI has lots of advantages, such as reducing parasitic capacitance, leakage current, power consumption, and enhancing the operating speed. It can also eliminate latch-up effects and suppress the interference of substrate.

Regarding the technical issues of further fabrication or integration with the sapphire wafer, in fact it is not a big problem because all the subsequent device processing steps are compatible with the current CMOS technologies. Moreover, the sapphire substrate can be processed at a high temperature of more than 1000 °C. The only issue is that the sapphire substrates are more expensive than the Si wafer, but the cost of sapphire is now decreasing, and a 6-inch wafer is already widely available for industry applications such as GaN technology.

Related references:

[54] Yoshimi, M. et al. Advantages of SOI technology in low-voltage ULSIs. *Microelectronic Manufacturing International Society for Optics and Photonics*, 1997.

[55] Ketchen, M. B. Competitive advantage of SOI from dynamic threshold shifts and reduced capacitance. *International Symposium on Vlsi Technology Systems & Applications Proceedings* (2003):129-132.

Q7: All measurement of various logic circuit units are plotted in the time scale of seconds. It would be ideal to show high frequency output characteristics.

Reply to the reviewer:

We thank the reviewer for the constructive suggestion.

Actually, there is a technical problem with high-frequency measurement. Compared with the impedance of a regular oscilloscope, the equivalent output impedance of our MoS₂ pseudo-NMOS inverter is too high to allow the detection of the output signal through an oscilloscope. Thus, a standard oscilloscope measurement is not possible, and more professional RF measurement facilities are required. However, in our laboratory, only a standard Agilent B1500A semiconductor analyzer is available to measure the output signal of our MoS₂ circuit. Based on Agilent B1500A, one can only apply a second-scale input signal to verify the logic function of the circuit, which makes it challenging to test higher frequency. Therefore, most previous literature results reported a low working frequency in large-scale MoS₂ circuits [56-58], and the RC delay has been discussed in answer to Q4.

Related references:

[56] Yu, L. et al. Design, Modeling, and Fabrication of Chemical Vapor Deposition Grown MoS₂ Circuits with E-Mode FETs for Large-Area Electronics. *Nano Letters* **16**, 6349-6356 (2016).

[57] Li, N. et al. Large-scale flexible and transparent electronics based on monolayer molybdenum disulfide field-effect transistors. *Nature Electronics* **3**, 711-717 (2020).

[58] Wang, L. et al. 2D Electronics: Electronic Devices and Circuits Based on Wafer-Scale Polycrystalline Monolayer MoS₂ by Chemical Vapor Deposition. *Advanced Electronic Materials* **5**, 1970038 (2019).

[59] Liu, Y. et al. Promises and prospects of two-dimensional transistors. *Nature* **591**, 43-53 (2021).

Q8: In fig. S17, photoresponse time of the device, such as raising and falling time, is in the scale of second. The performance might be a issue for real application. Is it due to interface issue or any

possible reasons.

Reply to the reviewer:

We thank the reviewer for the insightful suggestion. We acknowledge that the response speed of our photodetector based on 2D MoS₂ FETs is relatively slow, mainly due to the photogating effect [60], which is the dominant mechanism in our device.

Under illumination, free electron-hole pairs are generated. The trap states are then occupied by positively charged holes and act as a localized floating gate strongly modulating the channel conductance. Therefore, under illumination, the n-type transfer curves of MoS₂ FET are horizontally left-shifted compared with the dark state. Due to the slow de-trapping process, the long lifetime of the photo-generated carriers results in high gain but slow response speed.

The response speed of the MoS₂ photodetector can be further improved by increase the crystalline quality of the MoS₂, modifying the device structure, or apply a suitable gate voltage.

Related references:

[60] Fang, H. & Hu, W. Photogating in Low Dimensional Photodetectors. *Advanced Science* **4**, 1700323 (2017).

Q9: In fig. 3, the optimized performances are demonstrated with specific aspect ratio. This design of the device might raise issue on the speed or operations. Please explain this issue.

Reply to the reviewer:

We thank the reviewer for the constructive suggestion.

Indeed, we designed MoS₂ FETs with different aspect ratios for fabricating inverters with the V_M approach $V_{DD}/2$, which can enable a large noise margin and successful cascaded transmission of the voltage signal [61,62].

In the normal integrated circuit design process, another indispensable task is to determine the

aspect ratio of each transistor in addition to the design of circuit configuration in order to make the function, speed, power consumption and other indicators satisfy the design requirements of integrated circuits [63,64]. Therefore, it is appropriate to adjust the aspect ratio to ensure the circuit work properly. In order to shift V_M towards the positive voltage, it is necessary to appropriately increase the aspect ratio of the pull-up transistor to reduce the equivalent impedance of the pull-up network without changing the pull-down transistor.

Meanwhile, the gate capacitance and the parasitic capacitance will inevitably increase with the larger area of the pull-up transistor, which will slow down the operating speed of the circuits according to the equation $f_T = \frac{g_m}{2\pi C_G}$. In the future, different metals with various work functions can be adopted as the top gate electrode to adjust the V_T of the transistor, so that the pull-up transistor becomes a depletion-type device, the resistance of which is much smaller than that of the enhanced-type device at a particular voltage [65]. In this way, the equivalent impedance of the pull-up network can be reduced without increasing the size of the device, which is beneficial to improve the operating speed and power consumption of the integrated circuits.

Related references:

- [61] Yu, L. et al. in *2015 IEEE International Electron Devices Meeting (IEDM)*. 32.33.31-32.33.34.
- [62] Yang, Y., Ding, L., Han, J., Zhang, Z. & Peng, L.-M. High-Performance Complementary Transistors and Medium-Scale Integrated Circuits Based on Carbon Nanotube Thin Films. *ACS Nano* **11**, 4124-4132 (2017).
- [63] Wachter, S., Polyushkin, D. K., Bethge, O. & Mueller, T. A microprocessor based on a two-dimensional semiconductor. *Nature Communications* **8**, 14948 (2017).
- [64] Polyushkin, D. K. et al. Analogue two-dimensional semiconductor electronics. *Nature Electronics* **3**, 486-491 (2020).
- [65] Wang, H. et al. Integrated Circuits Based on Bilayer MoS₂ Transistors. *Nano Letters* **12**, 4674-4680 (2012).

Q10: In most reported papers on the grown MoS₂, overall performances are usually determined with many issues, such as grain size, interface, crystallinity, defect density, variation in the batch synthesis and more process details. The issues might be highly coupled. It might be a bit difficult for readers to understand how machine learning could work for the optimization.

Reply to the reviewer:

We thank the reviewer for the insightful suggestion.

We completely agree with the referee that the material characteristics of wafer-scale MoS₂, such as crystallinity, grain size, grain boundary property, defect density, etc., play an important role in device performance. These material characteristics depend on the synthesis recipe of MoS₂, which involves precursor types, growth temperature, carrier gas and other factors. Thus, if the material synthesis is coupled with the device processing, the burden of experimental work will be much increased and beyond the capability of a research lab. So, in this work, the optimization of material synthesis is not the focus of this paper. The synthesis recipe adopted in this paper has been investigated in detail before [66], which is relatively stable and is capable of synthesizing large-area films with high quality.

Therefore, the focus of this work is to optimize the fabrication process efficiently based on a fixed material. Actually, the ML method is quite like a black box, which is the working principle of ML (simply speaking, a statistical classification algorithm for large data sets), and people have already applied ML methods to assist the searching of low-dimensional materials [67]. In our case, ML represents a computer-aided learning process from large amounts of device data and human experience of device optimization.

Though ML represents a computer-aided learning process from large amounts of device data, the optimization process with ML also reveals some physical mechanisms behind the experimental data. For example, in Table S6 in the supplementary information, the optimum combination process obtained by ML analysis is the combination *c*, in which both contact (Ti/Au) and seeding layer (SiO₂) are not the best options obtained by single-step optimization. It can be partially explained by: 1) the addition of Ti as a buffer layer between MoS₂ film and Au electrodes

can effectively increase the adhesion of contacts, and its slightly smaller work function is beneficial to reduce the contact resistance and increase the on-state current [39]. 2) 2-nm-thick SiO₂ can reduce the damage to MoS₂ by the growth of HfO₂, but its defect level is also high because of the physical vapor deposition method. However, the subsequent annealing can likely repair the oxygen defects in the SiO₂ layer, thus substantially improving the quality of the dielectric layer and the transistor's electrostatic control capability [30].

Table R2. Comparison group with 3 different recipe combinations.

Process combination	S/D	Seed layer	Anneal of SL	Material of TG
a	Au	2 nm Al ₂ O ₃	W/	Au
b	Au	2 nm SiO ₂	W/	Au
c	Ti/Au	2 nm SiO ₂	W/	Au

Although understanding the underlying physics is not the key to this work, we believe our work opens the door for future detailed investigation of applying ML to optimize device fabrication. It also leaves open questions for researchers interested in the device physics of 2D semiconductors.

Related references:

[66] Xu, H. et al. High-Performance Wafer-Scale MoS₂ Transistors toward Practical Application. *Small* **14**, 1803465 (2018).

[67] Butler, K. T., Davies, D. W., Cartwright, H., Isayev, O. & Walsh, A. Machine learning for molecular and materials science. *Nature* **559**, 547-555 (2018).

[68] Liu, H. et al. Switching Mechanism in Single-Layer Molybdenum Disulfide Transistors: An Insight into Current Flow across Schottky Barriers. *ACS Nano* **8**, 1031-1038 (2014).

[69] Sheng, Y. et al. Gate Stack Engineering in MoS₂ Field-Effect Transistor for Reduced Channel Doping and Hysteresis Effect. *Advanced Electronic Materials* 2000395 (2020).

Q11: It is necessary to include detailed information on the fabricated devices, such as length/width and geometry of the FETs, material and size of the seeding layers, and thickness of the top dielectric.

Reply to the reviewer:

We thank the reviewer for the beneficial suggestion.

According to this suggestion, we provide a complete version of the optimized fabrication process in the supplementary information, shown below. We hope this can be helpful for other researchers to repeat our results.

Contact electrodes: Patterned by photolithography and deposited using Electronic Beam (E-beam) evaporation. Thickness: 10 nm Ti (sticking layer) and 50 nm Au. Evaporation rate: 0.1 Å/s for Ti and 1.0 Å/s for Au. Before evaporation sample was vacuum annealed in an E-beam evaporator at 150 °C for 1 hour.

MoS₂ channel formation: CF₄ plasma etching in an ICP (inductively coupled plasma) facility. After etching, the sample was immediately transferred to the E-beam evaporator and annealed at 150 °C for 1 hour for the subsequent seeding layer deposition.

Seeding layer: 2-nm-thick SiO₂ seeding layer was deposited by E-beam evaporation with an evaporation rate <0.1 Å/s. Then the sample was immediately transferred into a tube furnace and annealed in an oxygen atmosphere at 100 °C.

High-k dielectric layer: 20-nm-thick HfO₂ layer grown by atomic layer deposition (ALD) at 180 °C, where [Hf(N(CH₃)₂)₄] and H₂O are used as the precursors and N₂ as the carrier gas. After ALD growth, the sample was annealed in an oxygen atmosphere at 150 °C for 1 hour.

Top gate: 40-nm-thick Au by E-beam evaporation with an evaporation rate <0.5 Å/s. After the TG deposition, rapid thermal annealing (RTA) was performed at 150 °C for 5 min.

REVIEWER COMMENTS

Reviewer #1 (Remarks to the Author):

Dear authors,

Thank you for your detailed response to the review. I have carefully read through the rebuttal later and can confirm that I am completely satisfied with the response.

Best regards,
Dmitry Polyushkin.

Reviewer #2 (Remarks to the Author):

The authors have addressed the points raised by the reviewers in great detail. Some minor issues remain as indicated below. However, in my opinion, the merits of the machine learning approach still do not stand out, although I do understand the merits in industrial processing, where the amount of data is orders of magnitude larger and where miniscule improvements matter. The reason I am not convinced is rooted in the fact that the individual performance of the circuits, devices and their contributing components (electrical contacts, dielectric interfaces, hysteresis, equivalent oxide thickness, carrier mobility) is not particularly impressive. There is really no category in which the individual performance clearly outperforms the state-of-the-art (I am not comparing to demonstrators based on exfoliated crystals, but only devices and circuits based on large area grown MoS₂). Thus, I conclude that despite the effort spent using machine learning (50 person-years according to the authors), the same results could have been achieved with a conventional method, where one selects the metal of choice, dielectric of choice etc. based on careful design of experiment and literature study. This leaves the wafer scale fabrication approach as the main (and certainly impressive) achievement. In summary, while I am impressed by the broad demonstration of different devices and circuits, the individual components are not that impressive. Hence, I am not convinced by the manuscript that machine learning is the best approach towards demonstrating a fledgling technology.

There is one particular aspect that should be addressed independent of the statement above:

The authors have included important data on hysteresis. However, they only report a certain hysteresis voltage and do not provide details of the measurement. Since the root cause for hysteresis is quite complex and hysteresis depends on many measurement parameters, like sweep rate, hold time, maximum voltage, measurement environment and more, this information is required in detail.

Reviewer #3 (Remarks to the Author):

The manuscript has been revised and improved. The referee would like to recommend acceptance after some minor issues.

1

Regarding to Q10 (as mentioned in Q5/Q7 from reviewer #2/4), it would be ideal if the authors could include more discussions to explain how machine learning process and help on the

optimization of fabrication. It would be helpful to provide more insights compared to the 4 reported papers on fabrication of scalable 2D electronics.

2

In Fig S20, photoresponse of the devices on sapphire implies considerable trap states at interfaces. However, a ideal low freq $1/f$ noise is shown to demonstrate a reduced hysteresis with HfO₂/SiO₂ interface in Fig R2.

Reviewer #4 (Remarks to the Author):

I really appreciate the responses from the authors. All my comments have been properly addressed except Q1, which is about the advantage of ML over traditional DOE. In the response, the authors claims "Therefore, if the traditional design of experiment (DOE) method is adopted, a large number of combinations are needed for comparison, which dramatically increases the research workload and reduces the optimization work efficiency.", which implies that ML can save the number of combinations which is required for the process optimization. However, I am not convinced by this statement. I believe ML would even require more combinations in order to realize the same level of optimization from systematical DOE. Besides, the influence from the successive steps or the interference between different steps can also be address by systematical DOE. I disagree that this is something which can only be addressed by ML. I think this is a very important question to clarify before publication.

RESPONSE TO REVIEWERS

Reviewer #1 (Remarks to the Author):

Dear authors,

Thank you for your detailed response to the review. I have carefully read through the rebuttal later and can confirm that I am completely satisfied with the response.

Best regards,

Dmitry Polyushkin.

Reply to the reviewer:

We sincerely appreciate the reviewer for the elaborative reading of the reply letter and affirmation of our work. It is very encouraging to receive your positive comments on our work. Thanks again.

Reviewer #2 (Remarks to the Author):

The authors have addressed the points raised by the reviewers in great detail. Some minor issues remain as indicated below. However, in my opinion, the merits of the machine learning approach still do not stand out, although I do understand the merits in industrial processing, where the amount of data is orders of magnitude larger and where miniscule improvements matter. The reason I am not convinced is rooted in the fact that the individual performance of the circuits, devices and their contributing components (electrical contacts, dielectric interfaces, hysteresis, equivalent oxide thickness, carrier mobility) is not particularly impressive. There is really no category in which the individual performance clearly outperforms the state-of-the-art (I am not comparing to demonstrators based on exfoliated crystals, but only devices and circuits based on large area grown MoS₂). Thus, I conclude that despite the effort spent using machine learning (50 person-years according to the authors), the same results could have been achieved with a conventional method, where one selects the metal of choice, dielectric of choice etc. based on careful design of experiment and literature study. This leaves the wafer scale fabrication approach as the main (and certainly impressive) achievement. In summary, while I am impressed by the broad demonstration of different devices and circuits, the individual components are not that impressive. Hence, I am not convinced by the manuscript that machine learning is the best approach towards demonstrating a fledgling technology.

Reply to the reviewer:

We sincerely thank the reviewer for carefully reading our reply letter and revised manuscript. The additional insightful comments put forward by the reviewer are of great help to us. We also follow these comments to strengthen the discussion of machine learning in 2D device processing, and we hope that this version can qualify the publication standard of Nature Communications.

First, we entirely agree that the optimization can also be obtained by "*selection the metal of choice, dielectric of choice etc. based on careful design of experiment and literature study*". For example, people have exerted many efforts to achieve single-step optimizations, such as

compatible contact [1,2] and dielectric [3,4] recipes for 2D semiconductors. However, it is inconceivably tough to co-optimize the complete processing because all individual processing steps are highly coupled (as we explained in the previous reply letter, 2D semiconductors, especially those with small band-gap, are extremely sensitive to the exterior environments and fabrication processing, so any subsequent processing steps will influence the previous ones). This makes the process optimization of 2D semiconductors more complicated than those in bulk semiconductors such as Si and Ge. Thus, it inevitably requires numerous experiments for comparison and verification to comprehensively improve the device performance, especially under a conventional full-factorial design-of-experiment (DOE), which is highly time-consuming and labor-consuming. As far as we know, no experimental results have been reported before on a comprehensive optimization for 2D semiconductor device fabrication.

Moreover, unlike most previously reported results on large-scale 2D devices and circuits, the structure adopted in this paper is top-gate (TG) instead of their bottom-gate (BG) structures [5,7,8,10,11]. BG device architecture can avoid the TG doping problem, but it is inevitable to adopt a large-scale transfer technique, introducing extra impurities from the transfer tape and defects into 2D semiconductors. The uniformity also degrades after the transfer of 2D films, which is detrimental to the yield of large-scale circuits. Thus, it is unfair to compare our work with those based on BG-FETs. For TG structured 2D devices, it is difficult to perform a "step-by-step" optimization. For example, the contact fabrication that has already been optimized can still be influenced by the successive growth and annealing of the dielectric layer, as well as the top gate electrode deposition. Another example is that after the annealing of TG dielectric, not only the contact interface between the 2D semiconductor and the 3D metal electrodes is improved, it is also advantageous to repair the oxygen defects in the dielectric layer so that the interface between the 2D channel and the dielectric layer can also be improved.

Regarding device performance comparison, we also agree with the statement by the reviewer, "*There is really no category in which the individual performance clearly outperforms the state-of-the-art*". However, previous results focus only on individual factors,

such as mobility [8,9], threshold voltage [11], and subthreshold swing [4], etc. In our work, we tackle a more comprehensive optimization and wafer-scale device fabrication with a TG architecture, rather than aiming for individual “state-of-the-art” performance. As far as we know, no reported results show working TG structured enhance-mode 2D- FETs under a satisfactory wafer-scale uniformity. Most of the reported TG 2D-FETs with atomic layer deposition of high-k dielectric layer suffer from severe n-doping, limiting the cascading of large-scale circuits. It is also challenging to obtain a uniform and high-quality dielectric layer on wafer-scale 2D semiconductors, mainly because they lack dangling bonds on the surface for a homogeneous reaction.

Lastly, for the early development stage of devices based on emerging advanced materials, material quality variation is relatively large, hindering the following device optimization. Therefore, the light-weighted machine learning process used in this work can quickly locate the crucial aspects, and the predictive scoring and grid search were adopted to recommend possible working combinations. Combined with the knowledge and experience of the experts, the speed of device optimization could be significantly improved, which also converges the investigation efforts on device applications. So machine learning could be a powerful tool to assist researchers in reducing the investigation burden.

At the end of the supplementary material, we also compared the device performance and the circuit scale with previous works, as shown in Table R1. Our MoS₂ TG-FETs exhibit a satisfactory comprehensive performance, and the maximum transistor number in a functional circuit.

Area	V _{DS} (V)	I _{on} (A)	I _{off} (A)	I _{on} /I _{off}	Max. Mobility μ (cm ² V ⁻¹ s ⁻¹)	V _T (V)	Max. FETs number/IC	W/L	Gate type	Ref
~50mm ²	5	9×10 ⁻⁵	10 ⁻¹²	10 ⁸	~3	~-0.65	115	45/2	BG	5
2mm×3mm	2	2×10 ⁻⁵	10 ⁻¹¹	10 ⁶	3	~-1.3	3	45/3	TG	6
4 inch	3	10 ⁻³	10 ⁻¹³	10 ¹⁰	~55	~-1.7	12	30/6	BG	7
-	3	10 ⁻³	10 ⁻¹³	10 ¹⁰	~50	0.54	9	30/4	BG	8
1cm×0.5cm	1	~10 ⁻⁵	10 ⁻¹⁴	10 ⁸	> 40	-2	3	1/1	TG	9
-	1.5	10 ⁻⁵	10 ⁻¹⁴	10 ⁹	80	2.41	10	30/4	BG	10
5mm×5mm	8	12×10 ⁻⁵	10 ⁻¹⁴	10 ¹⁰	~20	~-3.2	12	4	BG	11
2 inch	0.5	5.65×10⁻⁵	10⁻¹⁴	10⁹	~88	2.47	156	30/20	TG	This work

Table R1. Comparison of MoS₂ FET performance with recently published results

Fig. R1. Transfer characteristics (a-g) for MoS₂ FETs reported in recently literature [5-11], respectively. (h) is a summary of the V_T and mobility of these results and ours. Notice that most results were based on BG structure.

Related references:

- [1] Liu, Y. *et al.* Approaching the Schottky–Mott limit in van der Waals metal–semiconductor junctions. *Nature* **557**, 696-700 (2018).
- [2] Shen, P.-C. *et al.* Ultralow contact resistance between semimetal and monolayer semiconductors. *Nature* **593**, 211-217 (2021).
- [3] Yu, Z. *et al.* in *2020 IEEE International Electron Devices Meeting (IEDM)*. 3.2.1-3.3.4.
- [4] Li, W. *et al.* Uniform and ultrathin high- κ gate dielectrics for two-dimensional electronic devices. *Nature Electronics* **2**, 563-571 (2019).
- [5] Wachter, S., Polyushkin, D. K., Bethge, O. & Mueller, T. A microprocessor based on a two-dimensional semiconductor. *Nature Communications* **8**, 14948 (2017).
- [6] Wang, L. *et al.* Electronic Devices and Circuits Based on Wafer-Scale Polycrystalline Monolayer MoS₂ by Chemical Vapor Deposition. *Advanced Electronic Materials* **5**, 1900393 (2019).
- [7] Li, N. *et al.* Large-scale flexible and transparent electronics based on monolayer

molybdenum disulfide field-effect transistors. *Nature Electronics* (2020).

[8] Yu, L. *et al.* in *International Electron Devices Meeting*. 32.33.31-32.33.34.

[9] Wang, H. *et al.* in *International Electron Devices Meeting*. 4.6.1-4.6.4.

[10] Yu, L. *et al.* Design, Modeling, and Fabrication of Chemical Vapor Deposition Grown MoS₂ Circuits with E-Mode FETs for Large-Area Electronics. *Nano Letters* **16**, 6349-6356 (2016).

[11] Polyushkin, D. K. *et al.* Analogue two-dimensional semiconductor electronics. *Nature Electronics* **3**, 486-491 (2020).

There is one particular aspect that should be addressed independent of the statement above: the authors have included important data on hysteresis. However, they only report a certain hysteresis voltage and do not provide details of the measurement. Since the root cause for hysteresis is quite complex and hysteresis depends on many measurement parameters, like sweep rate, hold time, maximum voltage, measurement environment and more, this information is required in detail.

Reply to the reviewer:

We thank the reviewer for the constructive suggestion. We also apologize for the lack of measurement details in the previous manuscript. As mentioned by the reviewer, the hysteresis originates from the complex interfaces and is influenced by sweep rate, hold time, maximum voltage, measurement environment, and other factors.

Our hysteresis data are obtained under identical measurement conditions, which facilitates a systematic comparison among the devices fabricated by different process combinations. The measurement details are provided in the revised manuscript:

“The measurement of transfer curves were all carried out at a gate sweep rate of 100mV/s (0.03V resolution), V_{TG} from -3 V to 3 V and hold time of 5s at the beginning and ending points, V_{DS} of 0.1 V, and under room temperature and atmospheric pressure. The

hysteresis voltage is determined by the V_{th} difference between the dual-sweep transfer characteristic curves.”

Reviewer #3 (Remarks to the Author):

The manuscript has been revised and improved. The referee would like to recommend acceptance after some minor issues.

Reply to the reviewer:

We sincerely appreciate the reviewer for carefully reviewing our revised manuscript and raising many insightful comments to make the article more readable. We also thank the reviewer for the recognition of our research.

1. Regarding to Q10 (as mentioned in Q5/Q7 from reviewer #2/4), it would be ideal if the authors could include more discussions to explain how machine learning process and help on the optimization of fabrication. It would be helpful to provide more insights compared to the 4 reported papers on fabrication of scalable 2D electronics.

Reply to the reviewer:

We thank the reviewer for the valuable suggestion. Indeed, it is essential to our work to “*explain how machine learning process and help on the optimization of fabrication*”, which was also raised by reviewer#2 again. We have also made a detailed reply to Reviewer #2 in this reply letter.

Simply speaking, when faced with large-volume parameters and experimental data, manual analysis is complicated and the workload is tremendous. To a large extent, the machine learning process can reduce the process exploration workload for device experts. Furthermore, based on the machine learning process and expert experience, the critical process parameters can be quickly located to search for the best process combination, significantly improving the research efficiency.

Regarding the details of the machine learning process, we added the following in the revised supplementary materials: “*The random forest algorithm can classify the sample set composed of different process combinations and device performance by using a weak classifier that can effectively process **discrete data** and evaluate the importance of each*

process step, which is mainly obtained through the Gini index evaluation. According to the evaluation result, several process steps such as the material quality, contact deposition, seeding layer, and TG deposition display the most obvious influence on the final performance of the device. All of them can be related to their corresponding physical mechanisms such as defects of MoS₂, Schottky barrier, interfacial scattering, work function and interface charge traps of TG metal. However, the detailed physical explanations are not the focus of this work. Then a score predictor based on the random forest algorithm was adopted to predict the results from all possible process combinations obtained by a grid search method, followed by realistic experiments which provide feedback and further improve the reliability of the score predictor.”

2. In Fig S20, photoresponse of the devices on sapphire implies considerable trap states at interfaces. However, a ideal low freq 1/f noise is shown to demonstrate a reduced hysteresis with HfO₂/SiO₂ interface in Fig R2.

Reply to the reviewer:

We thank the reviewer for the insightful suggestion.

Fig. S20 shows the time-resolved I_{PH} of the MoS₂ TG phototransistor, which exhibits a slow response speed mainly because the dominant mode of its photoelectric response is primarily based on the photogating effect, in which the charged traps occupied by photogenerated holes can act as a localized floating gate and result in electron doping. Due to the slow detrapping process, the long lifetime of the photogenerated carriers results in high gain but slow response speed, which is also related to the defects and band structure of MoS₂ itself [12,13].

The 1/f noise studies [14,15] have shown that near-interfacial oxide traps called "border traps" can exchange charge with the channel through carrier or emission by tunneling when the gate voltage is swept. The 1/f noise mainly represents the quality of the dielectric and the device reliability, from which we can also extract the border trap density, and the border

density is not the critical factor affecting the photoelectric response speed of the device.

Related references:

- [12] Fang, H. & Hu, W. Photogating in Low Dimensional Photodetectors. *Advanced Science* **4**, 1700323 (2017).
- [13] Long, M., Wang, P., Fang, H. & Hu, W. Progress, Challenges, and Opportunities for 2D Material Based Photodetectors. *Advanced Functional Materials* **29**, 1803807 (2019).
- [14] Fleetwood, D. M. et al. Effects of oxide traps, interface traps, and "border traps" on metal-oxide-semiconductor devices. *Journal of Applied Physics* **73**, 5058-5074 (1993).
- [15] Fleetwood, D. M. 'Border traps' in MOS devices. *IEEE Transactions on Nuclear Science* **39**, 269-271 (1992).

Reviewer #4 (Remarks to the Author):

I really appreciate the responses from the authors. All my comments have been properly addressed except Q1, which is about the advantage of ML over traditional DOE. In the response, the authors claims "Therefore, if the traditional design of experiment (DOE) method is adopted, a large number of combinations are needed for comparison, which dramatically increases the research workload and reduces the optimization work efficiency.", which implies that ML can save the number of combinations which is required for the process optimization. However, I am not convinced by this statement. I believe ML would even require more combinations in order to realize the same level of optimization from systematical DOE. Besides, the influence from the successive steps or the interference between different steps can also be address by systematical DOE. I disagree that this is something which can only be addressed by ML. I think this is a very important question to clarify before publication.

Reply to the reviewer:

We sincerely appreciate the reviewer for the careful reading of our reply letter and positive comments on our work. Indeed, the influence from the successive steps or the interference among different steps can also be addressed by conventional DOE. However, it requires much more labor costs and industrial investment. **Our method is more suitable for academic research groups.**

We also agree with the reviewer that experimental works are still necessary for the ML optimization method. Nevertheless, before starting this ML optimization project we have accumulated a large amount of experimental data during the last five years, most of the experiments were not guided by DOE in purpose, but these discrete data can still be used for training by the ML algorithm (a weak classifier is good at processing discrete data like fabricaton recipes). So this is a unique case for our research group.

In this work, the most significant merit for ML is for sorting all the possible processing combinations. A score predictor based on the random forest algorithm was adopted to predict the results from all possible process combinations obtained by a grid search method, followed by realistic experiments which provide feedback and further improve the reliability of the score predictor. In this case, our previously accumulated experimental results provide enough data for training such ML algorithms, which can effectively recommend possible working process combinations. It saves a lot of time to test all combinations because even for the traditional DOE, massive experiments for different process combinations must be designed and tested, which demands much more research workload and resources.

In the revised manuscript, we have deleted the statement about the "...if the traditional design of experiment (DOE) method is adopted, a large number of combinations are needed for comparison, which dramatically increases the research workload and reduces the optimization work efficiency.", and revised as "*Compared with traditional design of experiment (DOE) method, ML algorithm can effectively sort different process combinations, which dramatically reduces the research workload*".

We thank the referees again very much for carefully reading and offering valuable suggestions, which greatly improves our work.

REVIEWERS' COMMENTS

Reviewer #2 (Remarks to the Author):

Dear authors,

thank you for your patient response to my continued requests regarding the advantages (or not) of the Machine Learning approach. While I remain skeptical as to the advantages of ML over Design of Experiment, I acknowledge that ML may be an opportunity to speed up the initial phases in device optimization. Once a certain level is reached, however, I don't think that we can ignore the device Physics and defect analysis - we need to understand each aspect deeply in order to improve and optimize it.

I think that with the additional discussions added to the manuscript, it can now be published in Nature Communications. In particular, it is important to now read explicitly the explanation of the ML approach and its limits, including the benchmarking and the statement "However, the detailed physical explanations are not the focus of this work."

Reviewer #3 (Remarks to the Author):

This manuscript has been revised. The referee would like to recommend acceptance for publication.

Reviewer #4 (Remarks to the Author):

Thanks for the clarification from the authors. My only concern has been addressed in this revision.